# A sharp NMF result with applications in network modeling

**Jiashun Jin**
Department of Statistics & Data Science
Carnegie Mellon University
Pittsburgh, PA 15213
jiashun@stat.cmu.edu

## Abstract

Given an $n \times n$ non-negative rank-$K$ matrix $\Omega$ where $m$ eigenvalues are negative, when can we write $\Omega = ZPZ'$ for non-negative matrices $Z \in \mathbb{R}^{n,K}$ and $P \in \mathbb{R}^{K,K}$? While most existing works focused on the case of $m = 0$, our primary interest is on the case of general $m$. With new proof ideas, we present sharp results on when the NMF problem is solvable, which significantly extend existing results on this topic. The NMF problem is partially motivated by applications in network modeling. For a network with $K$ communities, rank-$K$ models are especially popular. The Degree-Corrected Mixed-Membership (DCMM) model is a recent rank-$K$ model which is especially useful and interpretable in practice. To enjoy such properties, it is of interest to study when a rank-$K$ model can be rewritten as a DCMM model. Using our NMF results, we show that for a rank-$K$ model in the most interesting parameter ranges, we can always rewrite it as a DCMM model.

## 1 Introduction

Fix $(n, K, m)$ where $n \geq K \geq 2$ and $0 \leq m \leq K - 1$. We are interested in the following *Non-negative Matrix Factorization (NMF)* problem.

The NMF problem: given an $n \times n$ symmetric non-negative irreducible matrix $\Omega$ with

rank $K$ where exactly $m$ of the $K$ nonzero eigenvalues are negative, when can we find (1.1)

non-negative matrices $Z \in \mathbb{R}^{n,K}$ and $P \in \mathbb{R}^{K,K}$ such that $\Omega = ZPZ'$?

**Definition 1.1** *We say a matrix $\Omega$ non-negative if all of its entries are non-negative, and we say it positive if all of its entries are (strictly) positive. We say the NMF problem is solvable for $\Omega$ is we can find non-negative matrices $(Z, P)$ as above such that $\Omega = ZPZ'$.*

We assume $K \geq 2$ for the case of $K = 1$ is trivial, and we assume $m \leq K - 1$ for an irreducible non-negative matrix has at least one positive eigenvalue (e.g., by Perron's theorem [9]).

NMF is a fundamental problem and has applications in areas such as image processing [5, 23], text learning [21], hyper-spectral unmixing, and social network analysis [13]. Our setting is a special case of NMF where both $\Omega$ and $P$ are symmetric, so we may call it *symmetric NMF*. In the literature, symmetric NMF was widely used in clustering of nonlinearly separable data from a similarity matrix [7], where for a non-negative symmetric matrix $\Omega$, it aims to find a non-negative matrix $Z$ such that

$$\Omega = ZZ', \qquad \text{where } Z \in \mathbb{R}^{n,N} \text{ and } N \geq K. \tag{1.2}$$

Note that, first, this implicitly requires that $\Omega$ is positive semi-definite. Second, it is understood that for many non-negative and positive semi-definite matrices $\Omega$, the smallest $N$ we can find in the

factorization of (1.2) is strictly larger than $K$ (the rank of $\Omega$). See the 2021 book by Shaked-Monderer and Berman [26]. The book is 551 pages and summarizes nicely most existing results on NMF.

Unfortunately, our setting in (1.1) is significantly different from that in (1.2), so existing results on NMF do not directly apply. Especially, our NMF setting is motivated applications of social network modeling, where we must (a) allow $\Omega$ to have negative eigenvalues, (b) require that $Z$ has exactly $K$ columns ($K = \text{rank}(\Omega)$), and (c) have a factorization of $\Omega = ZPZ'$ instead of $\Omega = ZZ'$ (we will soon see that both $(P, Z)$ have practical meanings in our setting).

Below, in Section 1.1, we introduce several recent network models. In Section 1.2, we explain why the NMF problem (1.1) is important and relevant in social network modeling.

## 1.1 Several recent rank-$K$ network models, and especially the DCMM model

Consider a symmetric connected network with $n$ nodes and let $A$ be the adjacency matrix, where $A(i, j) = 1$ if there is an edge connecting nodes $i$ and $j$ and $A(i, j) = 0$ otherwise. As a convention, we do not allow self edges, so all diagonal entries of $A$ are 0. We assume the network has $K$ perceivable communities (communities are scientifically meaningful but mathematically hard to define; intuitively, they are clusters of nodes that have more edges "within" than "across" [12, 30]): $\mathcal{C}_1, \mathcal{C}_2, \ldots, \mathcal{C}_K$. In many network models, we assume that the upper triangular entries of $A$ are independent Bernoulli random variables, and that there is an $n \times n$ non-negative matrix $\Omega$ such that $\Omega(i, j) = \mathbb{P}(A(i, j) = 1)$ for all $1 \le i \ne j \le n$. Let $\text{diag}(\Omega) \in \mathbb{R}^{n,n}$ be the diagonal matrix where the $i$-th diagonal entry is $\Omega(i, i)$ and let $W \in \mathbb{R}^{n,n}$ be the matrix where $W(i, j) = A(i, j) - \Omega(i, j)$ if $i \ne j$ and $W(i, j) = 0$ otherwise. The matrix $W$ is known as the generalized Wigner matrix. With these notations,
$$A = \Omega - \text{diag}(\Omega) + W. \tag{1.3}$$
We call $\Omega$ the *Bernoulli probability matrix*. Frequently, we assume a *rank-$K$* model for $\Omega$:

$$\Omega \text{ is an irreducible non-negative matrix where the rank is } K. \tag{1.4}$$

Note that $K$ is the number of communities and has important practical meanings. Also, irreducibility is a natural assumption as we assume the network is connected (otherwise, we can study each connected component of the network separately). Below are some examples of rank-$K$ models.

*Example 1 (RDPG Model).* In a Random Dot Product Graph (RDPG) model [28], we fix a $K$-dimensional distribution $F$, generate $y_i \overset{iid}{\sim} F$, and let $\Omega(i, j) = (y_i, y_j)$ (inner product), $1 \le i, j \le n$. If we write $Y = [y_1, y_2, \ldots, y_n]'$ (which is an $n \times K$ matrix), then $\Omega = YY'$. The model is well-known in network and graph modeling. However, a noteworthy issue is that, the matrix $\Omega$ defined in this way is always positive semi-definite. This makes the model relatively restrictive (e.g., [25]).

*Example 2 (GRDPG Model).* To address the issue above, Rubin-Delanchy *et al* [25] proposed the generalized RDPG (GRDPG). Fix $K$ and $0 \le m < K$. Let $J_{K,m} = \text{diag}(1, 1, \ldots, -1, \ldots, -1)$ be the $K \times K$ diagonal matrix where the first $(K - m)$ diagonal entries are 1 and the remaining diagonal entries are $-1$. With a similar $Y$ matrix as in RDPG, GRDPG assumes $\Omega = Y J_{K,m} Y'$. An $\Omega$ defined in this way has negative eigenvalues, but we have to choose $(Y, J_{K,m})$ carefully to make sure that $\Omega$ is non-negative; this problem is not immediately clear.

*Example 3.* It was argued (e.g., [4]) that the Bernoulli probability matrix $\Omega$ in a graphon model can be well-approximated by a low-rank matrix provided with some regularity conditions.

In all these examples above, the parameters do not have explicit practical meanings (at least not directly or not sufficiently), so in a real application example, it remains unclear how to interpret the estimates of these parameters. Therefore, it is desirable to have models where the parameters have more explicit meanings in practice and so are easier to interpret.

The Degree-Corrected Mixed-Membership (DCMM) model is one of such models. Proposed by [15] (see also [29]), the model is motivated by the observation that natural networks usually have severe degree heterogeneity and mixed-memberships. To accommodate both features, for each node $i$, $1 \le i \le n$, we use a (strictly positive) parameter $\theta_i$ to model the degree heterogeneity and a weight vector $\pi_i \in \mathbb{R}^K$ to model the memberships, where $\pi_i(k) =$ weight node $i$ puts in $\mathcal{C}_k$, $1 \le k \le K$. We call node $i$ pure if $\pi_i$ is degenerate (i.e., only one entry is nonzero) and mixed otherwise. We also model the community structure by a symmetric and non-negative matrix $P \in \mathbb{R}^{K,K}$:

$P(k, \ell) =$ baseline probability where a node in $\mathcal{C}_k$ and a node in $\mathcal{C}_\ell$ have an edge, $1 \le k, \ell \le K$.

DCMM assumes that for all $1 \leq i, j \leq n$, $\Omega(i, j) = \theta_i \theta_j \pi_i' P \pi_j$. If we let $\theta = (\theta_1, \ldots, \theta_n)'$, $\Pi = [\pi_1, \ldots, \pi_n]'$, and $\Theta$ be the $n \times n$ diagonal matrix where $\Theta(i, i) = \theta_i$, $1 \leq i \leq n$, then we have

$$\Omega = \Theta \Pi P \Pi' \Theta, \tag{1.5}$$

Conventionally, we assume $\text{rank}(\Pi) = \text{rank}(P) = K$, so DCMM is also a rank-$K$ model.

**Remark 1**. The DCMM model can be viewed as the extension of several models, including the classical block model. In fact, (a) DCMM reduces to *Degree-Corrected Block Model (DCBM)* [20] if all nodes are pure, (b) DCMM reduces to the *Mixed-Membership Stochastic Block Model (MMSBM)* [1, 2, 24] if all $\theta_i$ are equal, and (c) DCMM reduces to the classical *Stochastic Block Model (SBM)* [8] if all nodes are pure and all $\theta_i$ are equal (as above, node $i$ is pure if $\pi_i$ is degenerate).

## 1.2    When is a rank-$K$ network model also a DCMM model?

A DCMM model is a rank-$K$ model, but compared to other rank-$K$ models, all parameter matrices $(\Theta, \Pi, P)$ in the DCMM model have practical meanings and are easy to interpret. These make the DCMM model especially appealing in practice, and motivate the following problem:

$$\text{When is a rank-}K\text{ network model also a DCMM model?} \tag{1.6}$$

To explain why this is important, we use the dynamic co-citation networks in [11] (see also [10]) as an example. The paper presented 21 co-citation networks for the same set of nodes (i.e., authors) in statistics, each for a different time window. We are interested in (a) how many research areas in statistics, (b) what are baseline citation exchanges between different research areas, and (c) how the research interests of individual authors evolve over time. Here, a co-citation network is a symmetrized citation network where each node is an author, and two nodes have an edge if they have been co-cited for at least $N$ times (for an $N$ they picked) in the corresponding time window. The paper suggested that there are 3 primary research areas in statistics (which was interpreted as "Bayes", "Biostatistics", and "Non-parametric") and a handful of sub-areas, and that it is convenient to model each co-citation network by a DCMM model with $K = 3$. In detail, for each author $i$ and time window $t$, $1 \leq i \leq n, 1 \leq t \leq T$, they used a $K \times K$ matrix $P^{(t)}$ to model the baseline citation exchanges between the primary research areas, a positive number $\theta_{it}$ to model the relative influence (in citations) of author $i$, and a weight vector $\pi_{it}$ to model the research interest of author $i$. If we similarly let $\Theta^{(t)} = \text{diag}(\theta_{1t}, \theta_{2t}, \ldots, \theta_{nt})$ and $\Pi^{(t)} = [\pi_{1t}, \pi_{2t}, \ldots, \pi_{nt}]'$, then the Bernoulli probability matrix of the DCMM model at time $t$ is $\Omega^{(t)} = \Theta^{(t)} \Pi^{(t)} P^{(t)} (\Pi^{(t)})' \Theta^{(t)}$. Using the DCMM model, they discovered a research triangle of statisticians (reminiscent of Efron's triangle for statistical philosophy [6]), and used it to visualize the trajectories of research interests of a handful of individual authors.

Imagine that, if we use a different rank-$K$ model (e.g., GRDPG) to model these networks, say, with $\Omega^{(t)} = Y^{(t)} J^{(t)} (Y^{(t)})'$ for some matrices $(Y^{(t)}, J^{(t)})$. It is unclear how to relate $Y^{(t)}$ to baseline citation exchanges, research interests and relative influence of individual authors. This explains why (1.6) is of interest: given a rank-$K$ network model, we wish to know when we can rewrite it as DCMM model, and so we can enjoy the properties and interpretability of the DCMM model.

We now come back to (1.6). Seemingly, NMF is to key to answer this question. Consider a positive matrix $\Omega$ with rank $K$ and suppose that it has an NMF as in (1.1) for two non-negative matrices $Z \in \mathbb{R}^{n,K}$ and $P \in \mathbb{R}^{K,K}$: $\Omega = ZPZ'$. Write $Z = [z_1, z_2, \ldots, z_n]'$ so $z_i'$ is the $i$-th row. Without loss of generality, assume all $z_i$ are nonzero vectors. Let $\Theta(i, i) = \|z_i\|_1$ and $\pi_i = z_i / \|z_i\|_1$, $1 \leq i \leq n$. It is seen that $\Theta(i, i) > 0$, that each $\pi_i$ is a weight vector, and that $\Omega = ZPZ' = \Theta \Pi P \Pi' \Theta$. Therefore, we can always rewrite a rank-$K$ model as a DCMM model if $\Omega$ has an NMF as in (1.1). This explains our motivation underline the NMF problem (1.1).

Note that to answer the question in (1.1), a study on the NMF problem in (1.2) would be not be relevant. For example, in a DCMM model, $K$ is the number of communities, so an NMF in (1.2) with an $N > K$ would not be useful. For this reason, we have to focus on the NMF problem in (1.1).

## 1.3    Results and contributions

Write $\Omega = Y J_{K,m} Y'$ as in Example 2, where $J_{K,m} = \text{diag}(1, \ldots, 1, -1, \ldots, -1)$ is a $K \times K$ diagonal matrix and $Y = [y_1, y_2, \ldots, y_n]' \in \mathbb{R}^{n,K}$. Let $\lambda_k$ be the $k$-th eigenvalue of $\Omega$ and let $\xi_k$

be the corresponding eigenvector. For $1 \leq i \leq n$, define $r_i \in \mathbb{R}^{K-1}$ by $r_i(k) = \xi_{k+1}(i)/\xi_1(i)$, $1 \leq k \leq K - 1$. For any unit-norm vector $y_0 \in \mathbb{R}^K$, let $c(y_0) = \max_{\{1 \leq i \leq n\}}\{|(y_i, y_0)|/\|y_i\|\}$. In Section 2, we show that the NMF problem for $\Omega$ is solvable if $m \leq K/2$ and $c(y_0) \geq \sqrt{1 - 1/K}$ for some $y_0$; let us call this the main condition. We show that, in order for the NMF problem to be solvable, the constant $\sqrt{1 - 1/K}$ can not be further reduced. Therefore, in this sense, our results are sharp. Using this, we deduce several other results. Especially, we show that the NMF problem is solvable for $\Omega$ if $\sum_{k=1}^{K-1}(|\lambda_{k+1}| \cdot r_i^2(k)) \leq |\lambda_1|/(K-1)$ for all $1 \leq i \leq n$. We also extend our results to the case of $m > K/2$, and explain why we need a different proof in this case.

In Section 3, we apply our results on NMF to network modeling. We argue that for parameters in the most interesting range, we have (A) all $\|r_i\|$ are bounded, and (B) $\max_{2 \leq k \leq K}\{|\lambda_k/\lambda_1|\} \to 0$, and so the condition just mentioned holds. This implies that we can alway rewrite a rank-$K$ network model as a DCMM model if the parameters are in the most interesting range. We also discuss how to check the main condition in practice where $\Omega$ is unknown. We tackle this by proposing an approach to estimating $\Omega$, and support our results by some real networks.

Our contributions are two fold. First, we develop several new results on symmetric NMF (a problem of interest in many applications [5]). Existing works on symmetric NMF have been focused on the case of $m = 0$ (so $\Omega$ is positive semi-definite; $m$ is the number of negative eigenvalues of $\Omega$). In this case, the best result is seen to be [26, Theorem 3.137], which can be viewed as a special case of our results; see Remark 2. This suggests that our results are sharp, for they are hard to improve even in the special case of $m = 0$. Note that our case allows $m$ to take any possible values, so it is clearly harder to study. For example, to show the results for the case of $m = 0$, it suffices if we can find a $K \times K$ orthogonal matrix $Q$ such that $YQ'$ is non-negative, since $J_{K,m}$ is the identity matrix in this case. For our case, we must find a $Q$ such that $YQ'$ and $QJ_{K,m}Q'$ are simultaneously non-negative. Clearly, this requires new ideas. We tackle this by constructing a special class of matrices $Q$; see our proofs for details. Our approach is quite different from that of [26, Theorem 3.137] and is new.

Second, we shed interesting new light on different rank-$K$ network models. In the literature, it is not unusual that many similar models are proposed for the same type of data sets. But in the end, we need to understand the advantages and disadvantages of different models, and pick the most suitable one. Our study recommends DCMM model, for it offers desired practical interpretability which other rank-$K$ models do not have, and points out that a general rank-$K$ model is also a DCMM model if the parameters are in the most interesting range. Such findings are valuable for they can help us identify the most suitable models in real applications.

**Notations**. We denote $e_1, e_2, \ldots, e_K$ by the standard basis vectors of $K$-dimensional Euclidean space and $e_0 = K^{-1/2}(e_1 + e_2 + \ldots + e_K)$. For any $q > 0$ and vector $x$, $\|x\|_q$ denotes the $\ell^q$-norm (when $q = 2$, we drop the subscript and write $\|x\|$). For any two vectors $x$ and $y$ of the same dimension, $(x, y)$ denotes the inner product. For a vector $a \in \mathbb{R}^n$, $\mathrm{diag}(a)$ denotes the $n \times n$ diagonal matrix where the $i$-th diagonal entry is $a_i$, $1 \leq i \leq n$. When $\Omega$ is an $n \times n$ matrix, $\mathrm{diag}(\Omega)$ denotes the $n \times n$ diagonal matrix where the $i$-th entry is $\Omega(i,i)$, $1 \leq i \leq n$.

## 2  Main results on NMF

This section presents our results on NMF. Results on network modeling are in Section 3. Consider an $n \times n$ irreducible non-negative matrix $\Omega$ with rank $K$, where $n$ is usually much larger than $K$. By Perron's theorem [9], at least one eigenvalue of $\Omega$ is positive. Fix $0 \leq m \leq K - 1$ and suppose $\Omega$ has $m$ negative eigenvalues. Let $J_{K,m} = \mathrm{diag}(1, \ldots, 1, -1, \ldots, -1)$ be the $K \times K$ diagonal matrix as in Example 2. By basic algebra, we can always write

$$\Omega = Y J_{K,m} Y', \qquad \text{for a full rank matrix } Y \in \mathbb{R}^{n,K}. \tag{2.7}$$

We can also show (e.g., an exercise with the Weyl's theorem [9]) that for any matrix as in (2.7), the numbers of positive and negative eigenvalues are $(K - m)$ and $m$, respectively. Write

$$Y = [y_1, y_2, \ldots, y_n]', \qquad \text{so that } y_i' \text{ is row } i \text{ of } Y, 1 \leq i \leq n. \tag{2.8}$$

Define the subset of $K$-dimensional vectors that live on the unit-sphere where the last $m$ entries are 0:

$$\mathcal{S}_{K,m} = \{x = (x_1, \ldots, x_K)' \in \mathbb{R}^K, \|x\| = 1, x_{K-m+1} = \ldots = x_K = 0\}.$$

When $m = 0$, $\mathcal{S}_m$ is the unit sphere of $\mathbb{R}^K$. The following theorem is proved in the supplement.

**Theorem 2.1** *Fix $K \geq 2$, $n \geq K$, and $0 \leq m \leq K/2$. Consider the NMF problem (1.1) where $\Omega = Y J_{K,m} Y'$ and $Y$ are as in (2.7). Suppose there is a vector $y_0 \in \mathcal{S}_{K,m}$ such that*

$$|(y_0, y_i)|/\|y_i\| \geq \sqrt{1 - 1/K}, \qquad \text{for all } 1 \leq i \leq n. \tag{2.9}$$

*There exists a $K \times K$ orthogonal matrix $Q$ such that both $YQ'$ and $QJ_{K,m}Q'$ are non-negative. As a result, the NMF problem for $\Omega$ is solvable: $\Omega = ZPZ'$ with $Z = YQ'$ and $P = QJ_{K,m}Q'$.*

We have several comments. First, Theorem 2.1 assumes two conditions: $m \leq K/2$ and (2.9). When $K \leq 2$, both conditions hold automatically, so the NMF problem is always solvable in this case; see Section 2.1. As far as we know, our proof is different from existing approaches. Second, in Theorem 2.1, we require $y_0 \in \mathcal{S}_m$. This may seem restrictive, but is not. This is because $y_0$ is a vector we choose for our own convenience. In fact, one of the most interesting settings for NMF seems to be that in Section 2.3, where we choose $y_0 = (1, 0, \ldots, 0)'$, so the requirement is satisfied automatically. Also, when the last $m$ entries of $y_0$ are nonzero but sufficiently small, Theorem 2.1 continues to hold if we modify the term $\sqrt{1 - 1/K}$ slightly. Third, from a practical view point, the condition of $m \leq K/2$ is mild: we rarely see a rank-$K$ network model with $m > K/2$ (note here $m$ can be estimated using the eigenvalues of the adjacency matrix $A$). For theoretical completeness, the case of $m > K/2$ is also interesting, but there does not exist an orthogonal matrix $Q$ such that $QJ_{K,m}Q'$ is non-negative. This is because for any such $Q$, $\text{trace}(QJ_{K,m}Q') = K - 2m < 0$. Therefore, we must find a different way to solve the NMF problem in this case. We discuss this in Section 2.4. Last, an interesting question is whether our idea is extendable to asymmetric NMF or complex NMF [19]. As a simple extension to asymmetric NMF, consider an $n \times p$ positive matrix $\Omega$ of rank-$K$. By SVD, $\Omega = YZ'$ for an $n \times K$ matrix $Y$ and a $p \times K$ matrix $Z$. Let $y_i'$ be $i$-th row of $Y$ and $z_j'$ be the $j$-th row of Z, respectively. If there is a $y_0 \in \mathcal{S}_{K,m}$ such that for all $i$ and $j$, $|(y_i, y_0)|/\|y_i\| \geq \sqrt{1 - 1/K}$ and $|(z_j, y_0)|/\|z_j\| \geq \sqrt{1 - 1/K}$, then we can find a $K \times K$ orthogonal matrix $Q$ which rotates all rows of $Y$ and $Z$ to the first orthant simultaneously. In this case, the asymmetric NMF problem is solvable for $\Omega$. For reasons of space, we leave further study along this line to the future.

Our result is sharp for the constant $\sqrt{1 - 1/K}$ in (2.9) can not be further reduced. While we can show this for general $K$, we illustrate with the case of $K = 2$ for instruction purpose. In this case, we can rotate $n$ unit-norm vectors $y_1, y_2, \ldots y_n$ in $\mathbb{R}^2$ simultaneously to the first orthant if and only if there is a unit-norm vector $y_0$ such that $|(y_0, y_i)| \geq \sqrt{1 - 1/2}$ (i.e., the angle between them is $\leq \pi/4$) for all $1 \leq i \leq n$. See Section 2.1 and Remark 3 for more discussion. Another way to see the sharpness is to consider the case of $m = 0$ (so $\Omega$ is positive semi-definite). In this case, condition (2.9) is hard to improve and is the weakest we have so far in the literature; see Remark 2.

## 2.1 The case of $K = 2$

In this case, the NMF problem is always solvable, as the two conditions of Theorem 2.1, $m \leq K/2$ and (2.9), hold automatically. In fact, first, since $\Omega$ has at least one positive eigenvalues and $K = 2$, we have either $m = 0$ or $m = 1$, and so $m \leq K/2$. Second, we can always find a $y_0 \in \mathcal{S}_m$ such that (2.9) is satisfied. In detail, let $0 \leq \theta_i < 2\pi$ be the angle from $e_1$ ($e_1 = (1, 0)$) to $y_i$ counterclockwise, and let $\theta_{min}$ and $\theta_{max}$ be the smallest and largest values of all $\theta_i$. Now, when $m = 0$, let $y_0$ be the unit vector where the angle from $e_1$ to $y_0$ is $(\theta_{max} + \theta_{min})/2$, counterclockwise. When $m = 1$, take $y_0 = (1, 0)$. The following theorem is proved in the supplement.

**Theorem 2.2** *Fix $K = 2$, $0 \leq m \leq K - 1$, $n \geq K$, and let $y_0$ be as above. In this case, $m \leq K/2$ and (2.9) holds for the $y_0$ above, so the NMF problem is always solvable for $\Omega$.*

## 2.2 When $y_0$ is a scaled weighted average of $y_i$'s

For the $y_0$ in (2.9), an interesting choice is to let it be proportional to a weighted average of $y_i$'s. Call $w \in \mathbb{R}^n$ a weight vector if all of its entries are non-negative with a sum of 1. Recall that $\Omega = Y J_{K,m} Y'$. Define a proxy of $\Omega$ by $\widetilde{\Omega} = YY'$. Note that $\widetilde{\Omega} = \Omega$ if $m = 0$. Introduce $y^{(w)} \in \mathbb{R}^K$ and $\beta^{(w)} \in \mathbb{R}^n$ by $y^{(w)} = \sum_{i=1}^{n} w_i y_i = Y'w$ and $\beta^{(w)} = \widetilde{\Omega}w$. Since $Y$ is full rank, $y^{(w)} \neq 0$. Take $y_0 = y^{(w)}/\|y^{(w)}\|$. Condition (2.9) reduces to

$$|\beta_i^{(w)}|/\sqrt{\widetilde{\Omega}(i,i)(w'\widetilde{\Omega}w)} \geq \sqrt{1 - 1/K}, \qquad \text{for all } 1 \leq i \leq n. \tag{2.10}$$

**Theorem 2.3** *Fix $K \geq 3$, $0 \leq m \leq K/2$, and $n \geq K$. The NMF problem (1.1) is solvable for $\Omega$ if the last $m$ entries of $y^{(w)}$ are $0$ and (2.10) holds.*

Theorem 2.3 follows from Theorem 2.1 by direct calculations, so the proof is omitted. We require that the last $m$ entries of $y^{(w)}$ are $0$, for we need $y_0 \in \mathcal{S}_m$ in Theorem 2.1. As explained before, this may seem restrictive, but it is not, as in the most interesting case to be discussed in Section 2.3, we take $y^{(w)} = (1, 0, \ldots, 0)$, so the requirement is satisfied automatically. See details therein.

When $m = 0$, $\widetilde{\Omega} = \Omega$, and $\beta^{(w)} = \Omega w$. In this case, condition (2.10) reduces to

$$|\beta_i^{(w)}| / \sqrt{\Omega(i,i)(w'\Omega w)} \geq \sqrt{1 - 1/K}. \tag{2.11}$$

We have the following corollary, the proof of which is straightforwards so is omitted.

**Corollary 2.1** *Fix $n \geq K \geq 3$. The NMF problem (1.1) is solvable for $\Omega$ if $m = 0$ and (2.11) holds.*

**Remark 2**. If we take $w = n^{-1}\mathbf{1}_n$, then (2.10) reduces to $|\beta_i|/\sqrt{\Omega(i,i)(\mathbf{1}_n'\Omega \mathbf{1}_n)} \geq \sqrt{1 - 1/K}$ with $\beta = \Omega \mathbf{1}_n$, and Corollary 2.1 reduces to [26, Theorem 3.137], where $m = 0$ and $\Omega$ is positive semi-definite. Our setting is more general as $\Omega$ may have $m$ negative eigenvalues for any $m \leq K/2$. For the case of $m = 0$, [26, Theorem 3.137] (see also [27]) is by far the best results we can have. The book [26] presents several other results on this topic, but they need some conditions which are less intuitive or harder to check. Recall that the constant $\sqrt{1 - 1/K}$ in (2.9) can not be further reduced. These suggest that Theorem 2.1 is hard to improve and our results are sharp.

**Remark 3**. (*When can we rotate $n$ vectors to the first orthant?*) As a stylized application, consider the following problem. Let $x_1, x_2, \ldots, x_n$ be $n$ unit-norm vectors in $\mathbb{R}^K$, $n \geq K$, and let $\alpha_K(x_1, x_2, \ldots, x_n) = \min_{1 \leq i,j \leq n} \{(x_i, x_j)\}$. For what values of $\alpha_K(x_1, x_2, \ldots, x_n)$ can we rotate all $n$ points simultaneously to the first orthant? Let $X = [x_1, x_2, \ldots, x_n]'$ and assume $X$ is full rank without loss of generality. The matrix $\Omega = XX'$ is symmetric and positive semi-definite. Let $\alpha_K^* = 0$ if $K = 2$ and $\alpha_K^* = \sqrt{1 - 1/K}$ if $K \geq 3$. Applying Theorem 2.1 with $m = 0$, it follows that when $\alpha_K(x_1, x_2, \ldots, x_n) \geq \alpha_K^*$, we can rotate all $n$ points to the first orthant. Note that we can not do so if $\alpha_K(x_1, x_2, \ldots, x_n) < 0$.

## 2.3 When $Y$ is constructed by the spectral decomposition of $\Omega$

So far, we have tried to keep our results as general as we can, and $Y$ can be any matrix satisfying $\Omega = Y J_{K,m} Y'$. An interesting special case is when $Y$ is constructed using the spectral decomposition of $\Omega$, which we now discuss. For $1 \leq k \leq K$, let $\lambda_k$ be the $k$-th largest eigenvalue of $\Omega$, and let $\xi_k$ be the corresponding (unit-norm) eigenvector. In the literature $\lambda_1$ and $\xi_1$ are called the Perron root and Perron vector, respectively, where we can always assume all entries of $\xi_1$ are positive since $\Omega$ is irreducible and non-negative (e.g., [26]). Write $\Xi = [\xi_1, \xi_2, \ldots, \xi_K]$ and define the $n \times (K-1)$ so-called *matrix of entry-wise ratio* $R$ by $R(i,k) = \xi_{k+1}(i)/\xi_1(k)$, $1 \leq k \leq K - 1, 1 \leq i \leq n$ [12, 16]. Introduce

$$D = \mathrm{diag}(|\lambda_1|, |\lambda_2|, \ldots, |\lambda_K|), \qquad D_0 = \mathrm{diag}(|\lambda_2|, \ldots, |\lambda_K|), \tag{2.12}$$

and write

$$R = [r_1, r_2, \ldots, r_n]', \qquad Y = \Xi D^{1/2} = [y_1, y_2, \ldots, y_n]'. \tag{2.13}$$

By spectral decomposition, $\Omega = \Xi D^{1/2} J_{K,m} D^{1/2} \Xi' = Y J_{K,m} Y'$. Now, in Section 2.2, if we take $w = c\xi_1$ where $c = 1/\|\xi_1\|_1$, then by basic algebra and definitions, it is seen $y^{(w)} = c\sqrt{\lambda_1} e_1$ and so $y_0 = e_1$ and especially $y_0 \in \mathcal{S}_m$. Moreover, $\beta_i^{(w)} = c\lambda_1 \xi_1(i)$, $w'\widetilde{\Omega}w = c^2\lambda_1$, and $\widetilde{\Omega}(i,i) = y_i'Dy_i$. Combining these, condition (2.10) reduces to

$$r_i'D_0 r_i \equiv \sum_{k=1}^{K-1} (|\lambda_{k+1}| \cdot r_i^2(k)) \leq |\lambda_1|/(K-1), \qquad \text{for all } 1 \leq i \leq n. \tag{2.14}$$

The following theorem is proved in the supplement.

**Theorem 2.4** *Fix $K \geq 3$, $m \leq \frac{K}{2}$, and $n \geq K$. The NMF problem (1.1) is solvable if (2.14) holds.*

Note that as in most works on NMF (e.g., [26]), the main goal is to find *easy-to-check* conditions under which the NMF is solvable. Such conditions are sufficient but are not necessary.

## 2.4 The case of $m > K/2$

So far, we have been focused on the case of $m \leq K/2$, which is the case that is most frequently found in real networks. For completeness, we now consider the case where $m > K/2$. Since $0 \leq m \leq K-1$, such a case only exists when $K \geq 3$. In Theorem 2.1, we show that when $m \leq K/2$, we can find an orthogonal matrix $Q$ such that $QJ_{K,m}Q'$ is non-negative. When $m > K/2$, we can not do this, as for any such $Q$, $\text{trace}(QJ_{K,m}Q') = (K - 2m) < 0$. Therefore, we need a new approach. A convenient approach is to redefine $J_{K,m}$ where we select a subset of the positive diagonal entries of $J_{K,m}$ and add a positive number for each of them. Success has been shown in a related setting (e.g., [3]). Using such a trick, we can extend all our main results to the case of $m > K/2$. For reasons of space, we only consider an extension of Theorem 2.4, as the claim of the theorem is probably the most explicit. Also for reasons of space, we only consider the case where we add a number to the first diagonal entry of $J_{K,m}$. Note that the idea is readily extendable to more general cases.

Let $\mathcal{Q}$ be the set of all orthogonal matrices where the first column is $K^{-1/2}(1, 1, \ldots, 1)'$. Fix $1 \leq m \leq K - 1$. For any $Q \in \mathcal{Q}$, write $Q = [Q^{(K-m)}, Q^{(m)}]$, where $Q^{(K-m)}$ and $Q^{(m)}$ are the sub-matrix of $Q$ consisting the first $(K - m)$ columns and the other $m$ columns, respectively. Introduce a constant by $a_m = 1 + K \inf_{Q \in \mathcal{Q}} \max_{1 \leq i,j \leq K} \{H(i,j) : H = 2Q^{(m)}(Q^{(m)})' - I_K\}$ ($I_K$: $K \times K$ identity matrix). Theorem 2.5 extends Theorem 2.4 and is proved in the supplement.

**Theorem 2.5** *Fix $K \geq 3$, $0 \leq m \leq (K - 1)$, and $n \geq K$. We have $a_m = 1$ if $m \leq K/2$ and $a_m = (K - 1)$ if $m = K - 1$. Also, the NMF problem is solvable for $\Omega$ if $r_i'D_0r_i \equiv \sum_{k=1}^{K-1} |\lambda_{k+1}| r_i^2(k) \leq 1/[a_m(K - 1)]$ for all $1 \leq i \leq n$.*

When $m \leq K/2$, $a_m = 1$. In this case, the claim here is the same as that in Theorem 2.4.

**Remark 4**. When the NMF problem for $\Omega$ is solvable, the solution is usually not unique without a proper regularity condition (e.g., [5]). In our setting, once we can write $\Omega = \Theta\Pi P\Pi'\Theta$ for some non-negative matrices $(\Theta, \Pi, P)$ as in (1.5), the factorization is unique if (a) for each $1 \leq k \leq K$, there is at least one $i$ such that $\pi_j = e_k$, where $e_k$ is the $k$-th standard Euclidean basis vector of $\mathbb{R}^K$, and (b) all diagonal entries of $P$ are 1 (see [15, 16] for a proof).

**Remark 5**. When condition (2.9) of Theorem 2.1 holds for some vectors $y_0$, how to find such a $y_0$ and the orthogonal matrix $Q$ in Theorem 2.1 numerically? This is an interesting question and we discuss it in Section F of the supplement.

## 3 When is a rank-$K$ network model also a DCMM model

So far, we focus on general NMF settings where we show that the NMF problem (1.1) is solvable when, for example, (2.14) holds. We now apply the results to networks and study when we can rewrite a rank-$K$ network model as a DCMM model. Network analysis (e.g., community detection, membership estimation, link prediction) is a well-studied area, where we have a lot of knowledge on what is the regime of major interest and what conditions are reasonable [16, 15, 18, 29]. In fact, in network analysis, we usually use an asymptotic framework where $n \to \infty$, $K$ is fixed, and other parameters may vary with $n$, where it is quite acceptable to assume

(A) all $\|r_i\|$ are bounded $\quad$ and $\quad$ (B) $\max_{2 \leq k \leq K}\{|\lambda_k/\lambda_1|\} \to 0$;

the notations are the same as those in Theorem 2.4. In fact, (A)-(B) model the *most interesting regime* in network analysis. In Theorem 2.4, the main condition (e.g., (2.14)) is $r_i'D_0r_i \leq |\lambda_1|/(K - 1)$ for all $1 \leq i \leq n$. Once (A)-(B) hold, $(1/|\lambda_1|)D_0 \to 0$ and (2.14) holds, so we can always rewrite a rank-$K$ network model as a DCMM model when (A)-(B) hold.

The remaining question is then, why (A)-(B) are reasonable assumptions in network analysis, and why they model the most interesting regime in network analysis. We now explain these in details.

Let $\Omega$ be the Bernoulli probability matrix as in (1.3). Suppose $\Omega = YPY'$, where $Y \in \mathbb{R}^{n,K}$ and is full rank, $P \in \mathbb{R}^{K,K}$, and $(Y, P)$ are not necessarily non-negative. Denote $G = Y'Y$. Note that $G$ is a $K \times K$ symmetric and positive definite matrix. Let $G^{1/2}$ be the (unique) square root of $Y'Y$. We usually assume $Y$ is balanced in that (a) the $\ell^2$-norm of all $K$ columns are in the same order, and (b) no severe linearity between the $K$ columns [15, 18]. As a result, all eigenvalues of $G$ are at the same order. By basic algebra, there is a $K \times K$ orthogonal matrix $Q$ such that $\Xi = [\xi_1, \xi_2, \ldots, \xi_K] = YB$,

where $B = G^{-1/2}Q$. Write $B = [b_1, b_2, \ldots, b_K]$ and let $0 \leq \alpha_i < 2\pi$ be the angle between $b_1$ and $y_i$ (counterclockwise). Let $M(\Omega) = \max_{\{1 \leq i \leq n\}}\{1/|\cos(\alpha_i)|\}$ and define matrix $V \in \mathbb{R}^{K,K-1}$ by

$$V(i,k) = b_{k+1}(i)/b_1(i), \qquad 1 \leq i \leq K, \ 1 \leq k \leq K-1. \tag{3.15}$$

Write $V = [v_1, v_2, \ldots, v_K]'$ so $v_k'$ is row-$k$ of $V$, $1 \leq k \leq K$. For any symmetrical matrix $P$, $\lambda_k(P)$ denotes the $k$-th largest eigenvalue; to be consistent with earlier notations, we simply write $\lambda_k(\Omega)$ as $\lambda_k$. Lemma 3.1 is proved in the supplement.

**Lemma 3.1** *We have* $B = \mathrm{diag}(b_1)[\mathbf{1}_K, V]$, $P = B\mathrm{diag}(\lambda_1, \ldots, \lambda_K)B'$, $b_1$ *is an eigenvector of* $PG$, *and* $P(k,k) = b_1^2(k)[\lambda_1 + v_k'\mathrm{diag}(\lambda_2, \ldots, \lambda_K)v_k]$, $1 \leq k \leq K$. *Moreover, if as* $n \to \infty$, $\lambda_1(G) \leq c_0\lambda_K(G)$ *for a constant* $c_0 > 0$, *then condition (B) holds if and only if* $\max_{\{2 \leq k \leq K\}}\{|\lambda_k(P)/\lambda_1(P)|\} \to 0$, *and* $\max_{\{1 \leq i \leq n\}}\{\|r_i\|\} \leq CM(\Omega)$.

It is seen that conditions (A)-(B) hold if $M(\Omega) \leq C$ and $\max_{\{2 \leq k \leq K\}}\{|\lambda_k(P)/\lambda_1(P)|\} \to 0$. The first one is mild: it only requires that no $y_i$ is nearly orthogonal to $b_1$. To boil these conditions down to a more explicit and vivid form, we consider the DCMM model. It is fine to consider the DCMM model here for (a) we only use the model to explain why conditions (A)-(B) are reasonable, and (b) the argument below is extendable beyond the DCMM model. In the DCMM model, $\Omega = \Theta\Pi P\Pi'\Theta$. Therefore, we can write $\Omega = YPY'$ if we let $Y = \Theta\Pi$, where we note that $(Y, P)$ are non-negative. Recall that $G = Y'Y$ (a positive definite $K \times K$ matrix). Lemma 3.2 is proved in the supplement.

**Lemma 3.2** *If* $(Y, P)$ *are non-negative, then first,* $PG$ *is an irreducible non-negative matrix and* $b_1$ *is the Perron vector, so all entries of* $b_1$ *are strictly positive. Second, all rows of* $r_i$ *lives with a simplex with* $v_1, v_2, \ldots, v_K$ *being the vertices, so* $\max_{\{1 \leq i \leq n\}}\{\|r_i\|\} \leq \max_{\{1 \leq k \leq K\}}\{\|v_k\|\}$. *Last, if* $\lambda_1(G) \leq c_0\lambda_K(G)$, *then* $\max_{\{1 \leq i \leq n\}}\{\|r_i\|\} \leq CM(\Omega) \leq C\max_{1 \leq k \leq K}\{\|b_1\|/b_1(k)\}$.

Now, first, in a DCMM model, the matrix $P(k,\ell)$ measures the baseline probability where there is an edge between a node in community $k$ and a node in community $\ell$. Therefore, the *most difficult or most interesting case* is where all $P(k,\ell)$ have similar values. In this case, $P$ is close to rank-1, or in other words, $\max_{\{2 \leq k \leq K\}}\{|\lambda_k(P)/\lambda_1(P)|\} \to 0$, and so $\max_{\{2 \leq k \leq K\}}\{|\lambda_k/\lambda_1|\} \to 0$. See for example [15, 18], where it was further pointed out that the most difficult case for network analysis is when $\max_{\{2 \leq k \leq K\}}\{|\lambda_k|\} \leq L_n \cdot \sqrt{\lambda_1}$ for a multi-log$(n)$ factor $L_n$. Therefore, condition (A) models the most difficult case of network analysis and so is of major interest. Moreover, by Lemma 3.2, $\max_{\{1 \leq i \leq n\}}\{\|r_i\|\} \leq C$ if all entries of $b_1$ are at the same order. This is only a mild condition for $b_1$ is the Perron vector of $PG$. Last, by Lemma 3.2, we also have $\max_{\{1 \leq i \leq n\}}\{\|r_i\|\} \leq C$ if we alternatively assume $\max_{\{1 \leq k \leq K\}}\{\|v_k\|\} \leq C$. Recall that $B = G^{-1/2}Q = [b_1, b_2, \ldots, b_K]$ and $v_1', v_2', \ldots, v_K'$ are rows of $V$ formed by dividing $b_2, b_3, \ldots, b_K$ by $b_1$ entry-wise, where $b_1$ is the Perron vector. Since $G$ is positive definite where all eigenvalues are at the same order, $Q$ is orthogonal, and $V$ is properly scaled (and all of them have small-sizes), it is only a mild condition to assume $\max_{\{1 \leq k \leq K\}}\{\|v_k\|\} \leq C$. These explain why conditions (A)-(B) are mild condition and they model the most challenging regime for network analysis.

## 4 Real data examples, and especially how to check condition (2.14)

Let $a_i = (1/|\lambda_1|)r_i'D_0r_i$, $1 \leq i \leq n$. Condition (2.14) can be rewritten as $a_i \leq 1/(K-1)$, for all $1 \leq i \leq n$. In applications, $\Omega$ is unknown, so it is unclear how to obtain $a_i$. A straightforward approach is to estimate $a_i$ with the eigenvalues and eigenvectors of the adjacency matrix $A$, but the estimates may be too noisy. We propose the following approach, which is inspired by Lemmas 3.1-3.2 and the recent Mixed-SCORE approach [16]. Let $(Y, V)$ be as above. Mixed-SCORE suggests an interesting idea for estimating $V$ and (a normalized version of) $Y$, denoted by $\Pi$; see details therein. Let $\hat{\lambda}_k$ be the $k$-th eigenvalue of $A$ and let $\hat{\xi}_k$ be the corresponding eigenvector. Write $\widehat{\Xi} = [\hat{\xi}_1, \hat{\xi}_2, \ldots, \hat{\xi}_K] = [\hat{z}_1, \hat{z}_2, \ldots, \hat{z}_n]'$, so $\hat{z}_i'$ is row-$i$ of $\widehat{\Xi}$. Our approach runs as follows.

- Apply Mixed-SCORE and obtain an estimate $(\widehat{V}, \widehat{\Pi})$ for $(V, \Pi)$. Let $\hat{v}_k'$ be row $k$ or $\widehat{V}$ and let $\hat{\pi}_i'$ be row $i$ of $\widehat{\Pi}$, $1 \leq k \leq K$, $1 \leq i \leq n$.
- Estimate $b_1$ by $\hat{b}_1$ where $\hat{b}_1(k) = [\hat{\lambda}_1 + \sum_{k=2}^K \hat{\lambda}_k\hat{v}_k'\mathrm{diag}(\hat{\lambda}_2, \ldots, \hat{\lambda}_K)\hat{v}_k]^{-1/2}$. Let $\widehat{B} = \mathrm{diag}(\hat{b}_1)[\mathbf{1}_K, \widehat{V}]$, and estimate $P$ by $\widehat{P} = \widehat{B}\mathrm{diag}(\hat{\lambda}_1, \hat{\lambda}_2, \ldots, \hat{\lambda}_K)\widehat{B}'$. Let $\hat{y}_i = (\|z_i\|_1/\|\widehat{B}'\hat{\pi}_i\|_1)\hat{\pi}_i$, $1 \leq i \leq n$, and let $\widehat{Y} = [\hat{y}_1, \hat{y}_2, \ldots, \hat{y}_n]'$.

- Let $\hat{\mu}_k$ be the $k$-th eigenvalue of the matrix $\widehat{\Omega} = \widehat{Y}\widehat{P}\widehat{Y}'$, and let $\hat{\eta}_k$ be the corresponding eigenvector. In the definition of $a_i$ (see above and (2.14)), replace $(\lambda_k, \xi_k)$ by $(\hat{\mu}_k, \hat{\eta}_k)$ and denote the resultant quantity $\hat{a}_i$, $1 \leq i \leq n$. These are our estimates for $a_i$.

The approach can be shown to be consistent for $\Omega$ under some regularity conditions. We skip the study for it is beyond the scope of this paper. In this algorithm, $(\widehat{Y}, \widehat{P})$ are not automatically non-negative, and to check whether NMF is solvable for $\widehat{\Omega}$, we can check if

$$\hat{a}_i \leq 1/(K-1), \qquad \text{for all } 1 \leq i \leq n. \tag{4.16}$$

**Remark 6**. Condition (2.14) of Theorem 2.4 is only a sufficient condition for NMF; they are not necessary conditions. It could happen that an NMF is solvable for an $\Omega$ but (2.14) does not hold.

We now consider some real examples. The weblog is a well-known data set [22], where with some light preprocessing, the network has $1,222$ node (each is a blog) and $16,714$ edges (each is a two-way hyperlink). The network has two communities: democratic and republican. For this data set, a rank-2 model is appropriate, so we have $(n, K) = (1,222, 2)$ (e.g., [30, 12, 18]). Let $\Omega$ be the Bernoulli probability matrix as in (1.3). By Theorem 2.2, when $K = 2$, we can always decompose $\Omega$ as $\Omega = YPY'$ for a non-negative $n \times 2$ matrix $Y$ and a $2 \times 2$ non-negative matrix $P$. Now, by the paragraph right above Remark 1, we can rewrite $\Omega = \Theta\Pi P\Pi\Theta$ as in (1.3), so $\Omega$ satisfies a DCMM model. Same claim can be drawn for the karate data set [30, 12], where we similarly have $K = 2$.

As another example, we consider the UKFaculty network (e.g., see [17, Table 1]). It is reasonable to model the network with a rank-$K$ model with $(n, K) = (81, 3)$ and $m \leq K/2$. By Theorem 2.4, the model can be rewritten as a DCMM model if (4.16) holds. Following the discussion above, we first obtain an estimate $\widehat{\Omega}$ for $\Omega$. We then use $\widehat{\Omega}$ to obtain $\hat{a}_i$ and check if (4.16) holds. The results are in Figure 1 (left) below, where the maximum of $\hat{a}_1, \hat{a}_2, \ldots, \hat{a}_n$ is slightly smaller than $0.5$ ($1/(K-1) = 0.5$ as $K = 3$), suggesting that (4.16) holds. Moreover, let $\hat{\mu}_k$ be the $k$-th eigenvector of $\widehat{\Omega}$ and let $\hat{\eta}_k$ be the corresponding eigenvector. Let $\widehat{D} = \text{diag}(\hat{\mu}_1, \ldots, \hat{\mu}_K)$ and $\widehat{Y} = [\hat{\eta}_1, \ldots, \hat{\eta}_K]\widehat{D}^{1/2}$. We have $\widehat{\Omega} = \widehat{Y}J_{K,m}\widehat{Y}'$. Let $Q$ be the $3 \times 3$ matrix where the three rows are $(1/\sqrt{3}, 1/\sqrt{6}, 1/\sqrt{2})$, $(1/\sqrt{3}, 1/\sqrt{6}, -1/\sqrt{2})$, and $(1/\sqrt{3}, -2/\sqrt{6}, 0)$, respectively. Define $\widehat{Z} = \widehat{Y}Q'$. It is seen $\widehat{\Omega} = \widehat{Y}J_{K,m}\widehat{Y}' = \widehat{Z}[QJ_{K,m}Q']\widehat{Z}'$, where $QJ_{K,m}Q'$ is seen to be non-negative. Moreover, for $1 \leq i \leq n$, let $\hat{z}_i$ be the smallest entry in row-$i$ of $\widehat{Z}$. Figure 1 (right) plots the histogram for $\{\hat{z}_i\}_{i=1}^n$. The results suggest that all $\hat{z}_i$ are non-negative, so the matrix $\widehat{Y}Q'$ is non-negative. Therefore, $\widehat{\Omega}$ has an NFM by $\widehat{\Omega} = \widehat{Z}[QJ_{K,m}Q']\widehat{Z}'$. These suggest that for the UKFaculty data set, (4.16) holds and it is reasonable to model the UKFaculty with a DCMM model. In summary, in many recent works on network analysis, we frequently assume that a DCMM model holds for the settings at hand, but we rarely checked if such an assumption is valid. Our NMF results provide an approach to checking whether the network satisfies DCMM model.

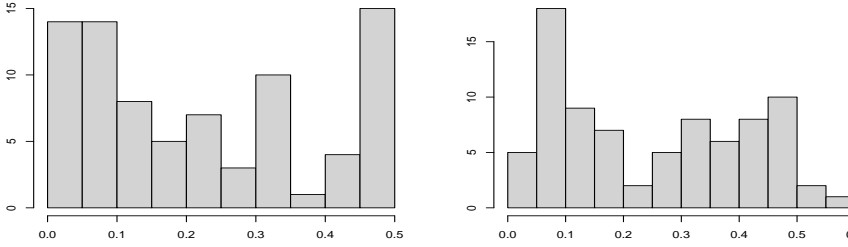

Figure 1: Histograms of $\{\hat{a}_i\}_{i=1}^n$ (left) and $\{\hat{z}_i\}_{i=1}^n$ (right). The results suggest that all $\hat{a}_i$ are smaller than $0.5$ (note that $1/(K-1) = .5$ as $K = 3$) so condition (4.16) is satisfied, and that all $\hat{z}_i$ are non-negative so the matrix $\widehat{Z}$ is non-negative. See above for definitions of $\hat{a}_i$ and $\hat{z}_i$.

# 5  Discussion

We derive a sharp NMF result and apply it to network modeling. Both NMF and network analysis are important areas in machine learning, with applications in image processing, social media, NLP, and cancer study [5, 23, 21]. In comparison, NMF is more theoretically oriented and network analysis is more application oriented. Our paper makes an interesting connection of the two areas. On one hand, we find a new application of NMF theory. This may open the door for a line of research where we find new applications of NMF in areas such as text learning [21] and tensor analysis [14]. On the other hand, we gain valuable insight on what are the most suitable network models in applications. This is crucial, for a suitable model is the starting point for methods and theory. Our study may help researchers identify the right network models and so can channel their strengths to the right direction. Our work may also help develop new methods. For example, compared to the general rank-$K$ model, the DCMM model has more structures which we can exploit (see [16, 18] where they discovered a simplex structure in the spectral domain, using some specific features which the DCMM model has but a general rank-$K$ model does not). Our approach is useful for it ensures us that in certain settings, we can use a more specific model and exploit the structures the model provides.

Another point is that, existing NMF theory usually requires some crucial conditions. However, whether such conditions are reasonable in real applications remains unclear, especially when the conditions are on matrices that are not directly observable. In Section 3-4, we tackle this problem by providing (a) a detailed explanation for why our NMF assumptions are reasonable in network analysis and (b) new ideas for checking the NMF conditions in real applications when the NMF conditions are on matrices that are not directly observable. We hope our efforts many spark some new research along this line.

**Acknowledgements**. The research was supported in part by NSF Grant DMS-2015469. The author would like to thank Naomi Shaked-Monderer, Helena Smigoc, and Changqing Xu for helpful pointers, and Zheng Tracy Ke and Jiajun Tang for very helpful comments.

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
