# Supplement of "A sharp NMF result with applications in network modeling"

**Jiashun Jin**
Department of Statistics & Data Science
Carnegie Mellon University
Pittsburgh, PA 15213
jiashun@stat.cmu.edu

In this supplement file, we prove the presented results. Note that Theorem 2.3 and Corollary 2.1 follow directly from Theorem 2.1 by basic algebra, so the proof are omitted. In this paper, $C$ is a generic constant that may vary from occurrence to occurrence.

## A  Proof of Theorem 2.1

The key is to prove the following lemma, which the conditions are similar to those of Theorem 2.1, except for that we replace condition (2.9) in Theorem 2.1 by a slightly stronger condition:

$$\frac{(y_0, y_i)}{\|y_i\|} \geq \sqrt{1 - 1/K}, \qquad \text{for all } 1 \leq i \leq n. \tag{A.1}$$

**Lemma A.1** *Fix $K \geq 2$, $n \geq K$, and $0 \leq m \leq K/2$. Consider the NMF problem as in (1.1), where $J_{K,m}$ and $Y$ are as in (2.7). Suppose there is a vector $y_0 \in \mathcal{S}_{K,m}$ such that*

$$\frac{(y_0, y_i)}{\|y_i\|} \geq \sqrt{1 - 1/K}, \qquad \text{for all } 1 \leq i \leq n.$$

*There is a $K \times K$ orthogonal matrix $Q$ such that simultaneously*

- *For all $1 \leq i \leq n$, $Qy_i$ falls in the first orthant of $\mathbb{R}^K$.*

- *$QJ_{K,m}Q'$ is non-negative.*

- *The NMF problem for $\Omega$ is solvable by writing*
$$Y J_{K,m} Y' = Z P Z' \text{ with } Z = YQ' \text{ and } P = Q J_{K,m} Q'.$$

Lemma A.1 is proved below.

We now explain why Theorem 2.1 follows once Lemma A.1 is proved. In Theorem 2.1, $\Omega = Y J_{K,m} Y'$, and $Y = [y_1, y_2, \ldots, y_n]'$. By the assumption of Theorem 2.1,

$$\frac{|(y_0, y_i)|}{\|y_i\|} \geq \sqrt{1 - 1/K}, \qquad 1 \leq i \leq n.$$

Therefore, for each $1 \leq i \leq n$, there is a number $s_i \in \{-1, 1\}$ such that

$$\frac{(y_0, s_i y_i)}{\|y_i\|} \geq \sqrt{1 - 1/K}.$$

We have two cases.

- Case 1. $s_i = -1$ for all $1 \leq i \leq n$.

- Case 2. $s_i = 1$ for at least one $1 \leq i \leq n$.

In case 1, let $\widetilde{Y} = -Y$. Theorem (2.1) follows directly if we write $\Omega = \widetilde{Y} J_{K,m} \widetilde{Y}$ and then apply Lemma A.1. In case 2, the key is to show that

$$\text{if } s_i = 1 \text{ for some } 1 \leq i \leq n, \text{ then } s_i = 1 \text{ for all } 1 \leq i \leq n. \tag{A.2}$$

Note that once (A.2) is proved, Theorem 2.1 follows by directly applying Lemma A.1 in this case.

It remains to show (A.2). Let $\widetilde{Y} = [\tilde{y}_1, \tilde{y}_2, \dots, \tilde{y}_n]'$ where $\tilde{y}_i = s_i y_i$ and let

$$\widetilde{\Omega} = \widetilde{Y} J_{K,m} \widetilde{Y}'.$$

It is seen that $\widetilde{\Omega}$ satisfies all conditions of Lemma A.1, so there is a $K \times K$ orthogonal matrix $Q$ such that $Q J_{K,m} Q'$ is non-negative, and that $Q \tilde{y}_i$ is non-negative for all $1 \leq i \leq n$.

Now, let

$$S = \{1 \leq i \leq n : s_i = 1\}.$$

If (A.2) is not true, $S^c$ is non-empty. Since $\Omega$ is irreducible, we can find an $i \in S$ and $j \notin S$ such that $\Omega(i,j) > 0$. Note that

$$\Omega(i,j) = y_i' J_{K,m} y_j = s_i s_j \tilde{y}_i J_{K,m} \tilde{y}_j = s_i s_j (Q \tilde{y}_i)' [Q J_{K,m} Q'](Q y_j). \tag{A.3}$$

Therefore, on one hand, $\Omega(i,j) > 0$. On the other hand, since $Q J_{K,m} Q'$, $Q \tilde{y}_i$, and $Q \tilde{y}_j$ are all non-negative, $(Q \tilde{y}_i)' [Q J_{K,m} Q'](Q y_j) \geq 0$. Note however $s_i s_j = -1$. Comparing these with (A.3) gives a contradiction. The contradiction shows that $S^c$ is empty and completes the proof.

## A.1 Proof of Lemma A.1

For $1 \leq k \leq K/2$, let $h_k$ be the vector where

$$h_k(i) = \begin{cases} 1/\sqrt{2}, & i = 2k - 1, \\ -1/\sqrt{2}, & i = 2k, \end{cases}$$

and let

$$Q^{(m)} = [h_1, h_2, \dots, h_m].$$

Note that in the special case of $m = 0$, $Q^{(m)}$ is empty.

We need the following lemma. As before, let $e_0$ be the $K$-dimensional vector $K^{-1/2}(1, \dots, 1)'$.

**Lemma A.2** *Any $K$-dimensional unit-norm vector $x$ with $(x, e_0) \geq \sqrt{1 - 1/K}$ is non-negative.*

**Proof of Lemma A.2.** Without of loss of generality, we assume the first $k$ entries of $x$ are non-negative, and the remaining $(K - k)$ entries are strictly negative. All we need to show is $k = K$. If $k < K$, then by definition and Cauchy-Schwartz inequality,

$$(x, e_0) < K^{-1/2} \sum_{i=1}^{k} x_i \leq K^{-1/2} \sqrt{k \sum_{k=1}^{k} x_i^2} < \sqrt{k/K},$$

where we have used

$$\sum_{k=1}^{k} x_i^2 < \|x\|^2 = 1.$$

This contradicts with $(x, e_0) \geq \sqrt{1 - 1/K}$, and so the claim follows.

We also need the following lemma.

**Lemma A.3** *Fix $K \geq 2$, $0 \leq m \leq K/2$, and a unit-norm vector $y_0 \in \mathcal{S}_m$. There exists a $K \times (K - m)$ matrix $B$ such that*

$$B'B = I_{K-m}, \qquad B'Q^{(m)} = 0, \qquad \text{and} \qquad [B, Q^{(m)}]'e_0 = y_0.$$

**Proof of Lemma A.3**. The proof for the case of $m = 0$ is trivial, so we only consider the case of $1 \le m \le K/2$. It is seen that $\{h_1, h_2, \ldots, h_m\}$ expands a $m$-dimensional sub-space of $\mathbb{R}^K$. By basic algebra, we can always expand $\{h_1, h_2, \ldots, h_m\}$ to a full set of orthogonal basis vectors of $\mathbb{R}^K$, denoted by

$$\{q_1, q_2, \ldots, q_{K-m}, h_1, h_2, \ldots, h_m\}.$$

Note that since $e_0 \perp \mathrm{span}\{h_1, h_2, \ldots, h_m\}$ (i.e., the linear space spanned by $h_1, \ldots, h_m$), there is a $(K - m)$ vector $\delta_1$ such that

$$e_0 = \sum_{i=1}^{K-m} \delta_i q_i = Q_0 \delta, \qquad \text{where } Q_0 = [q_1, q_2, \ldots, q_{K-m}].$$

Since $Q_0' Q^{(m)} = 0$ and $Q_0' Q_0 = I_{K-m}$,

$$\|\delta\| = \|Q_0 \delta\| = \|e_0\| = 1.$$

At the same time, since $y_0 \in \mathcal{S}_m$, by definition, there is a unit-norm $(K - m)$ dimensional vector $a$ such that

$$y_0 = (a, 0, \ldots, 0)',$$

and there is a $(K - m) \times (K - m)$ orthogonal matrix $U$ such that

$$U' \delta = a.$$

Let

$$B = Q_0 U.$$

We have

- $B'B = U'Q_0' Q_0 U = I_{K-m}$.
- $B'Q^{(m)} = U'Q_0 Q^{(m)} = 0$.
- $[B, Q^{(m)}]' e_0 = \begin{bmatrix} U'Q_0' Q_0 \delta \\ (Q^{(m)})' e_0 \end{bmatrix} = \begin{bmatrix} a \\ 0 \end{bmatrix} = y_0$.

This proves Lemma A.3.

We now come back to prove Lemma A.1. Let $B$ by any matrix in Lemma A.3, and let

$$Q = [B, Q^{(m)}].$$

By Lemma A.3,

$$Q' e_0 = y_0, \qquad \text{and so} \qquad Q y_0 = e_0.$$

At the same time, for all $1 \le i \le n$,

$$(y_i, y_0) = (Q y_i, Q y_0) = (Q y_i, e_0), \qquad \|y_i\| = \|Q y_i\|.$$

Combining these with (A.1),

$$\frac{(Q y_i, e_0)}{\|Q y_i\|} = \frac{(y_i, y_0)}{\|y_i\|} \ge \sqrt{1 - 1/K}.$$

It follows from Lemma A.2 that

$$\text{for all } 1 \le i \le n, \; Q y_i \text{ is a non-negative vector.} \tag{A.4}$$

At the same time, since $[B, Q^{(m)}]$ is an orthogonal matrix, $BB' + Q^{(m)}(Q^{(m)})' = I_K$, and so

$$Q J_{K,m} Q' = BB' - Q^{(m)}(Q^{(m)})' = I_K - 2Q^{(m)}(Q^{(m)})'.$$

By direct calculations, $2Q^{(m)}(Q^{(m)})'$ is a $K \times K$ block-wise diagonal matrix, where the first $m$ diagonal blocks are

$$\begin{bmatrix} 1 & -1 \\ -1 & 1 \end{bmatrix};$$

note that except for these $m$ diagonal blocks, the matrix is $0$ elsewhere. Therefore,

$$Q J_{K,m} Q' = I_K - 2Q^{(m)}(Q^{(m)})' \text{ and is non-negative.} \tag{A.5}$$

Combining this with (A.4)-(A.5) gives Lemma A.1.

# B Proof of Theorem 2.2

By Theorem 2.1, we only need to check when $K = 2$,

- we must have $m \leq K/2$.
- the condition (2.9) always holds without any extra conditions.

Since $\Omega$ has at least one positive eigenvalues, so we can only have $m = 0$ or $m = 1$. Therefore, we must have $0 \leq m \leq K/2$. This checks the first bullet point above.

We now consider the second bullet point. In detail, let $e_1 = (1, 0)'$ as before. Let $0 \leq \theta_i < 2\pi$ be the angle from $e_1$ to $y_i$ counterclockwise, and let $\theta_{min}$ and $\theta_{max}$ be the smallest and largest values of all $\theta_i$. Recall that we can only have $m = 0$ or $m = 1$. Also, we assume the following in the lemma.

- When $m = 0$, $y_0$ is the unit vector where the angle from $e_1$ to $y_0$ is $(\theta_{max} + \theta_{min})/2$, counterclockwise.
- When $m = 1$, $y_0 = (1, 0)$.

We now consider the case of $m = 0$ and $m = 1$ separately.

Consider the case of $m = 0$ first. Note that all $y_i$ fall in the sector (with the apex at 0) bounded by two rays, where the angle from $e_1$ to the two rays (counterclockwise) are $\theta_{min}$ and $\theta_{max}$, respectively. Recall that $\Omega = Y J_{K,m} Y'$. When $m = 0$, $J_{K,m} = I_K$ and $\Omega = YY'$. Therefore, for $1 \leq i, j \leq n$,

$$(y_i, y_j) = \Omega(i, j) \geq 0.$$

This says the angle of the sector is no bigger than $\pi/2$:

$$\theta_{max} - \theta_{min} \leq \pi/2.$$

If we take $y_0$ as above (so the angle between $e_1$ and $y_0$ is $(\theta_{max} + \theta_{min})/2$), it follows that for each $1 \leq i \leq n$, the angle between $y_0$ and $y_i$ is no bigger than $\pi/4$. Therefore,

$$\frac{|(y_i, y_0)|}{\|y_i\|} \geq \sqrt{1/2},$$

and (2.9) holds.

Consider the case of $m = 1$. In this case,

$$J_{K,m} = \begin{bmatrix} 1 & 0 \\ 0 & -1 \end{bmatrix}.$$

For each $1 \leq i \leq n$, write $y_i = (a_i, b_i)'$. Since $\Omega = Y J_{K,m} Y'$ and $\Omega(i, i) > 0$,

$$0 \leq \Omega(i, i) = y_i' J_{K,m} y_i = a_i^2 - b_i^2,$$

and so

$$|b_i| \leq |a_i|.$$

If we take $y_0 = e_1$ as above, then for all $1 \leq i \leq n$.

$$\frac{|(y_i, y_0)|}{\|y_i\|} = \frac{|a_i|}{\sqrt{a_i^2 + b_i^2}} \geq \sqrt{1/2},$$

and the claim follows directly.

# C Proof of Theorem 2.4

It is sufficient to justify

- $y_0 = e_1$ (and so especially $y_0 \in \mathcal{S}_m$).
- condition (2.10) is equivalent to

$$\sum_{k=1}^{K-1} |\lambda_{k+1}| \cdot r_i^2(k) \leq \lambda_1/(K-1), \tag{C.6}$$

where $r_i(k) = \xi_{k+1}(i)/\xi_1(i)$, $1 \leq k \leq K - 1$, $1 \leq i \leq n$.

We now justify both bullet points.

Consider the first bullet point first. Note that in this case,

$$\Omega = Y J_{K,m} Y', \qquad Y = [\xi_1, \xi_2, \ldots, \xi_K] D^{1/2},$$

and

$$y^{(w)} = Y' w,$$

with

$$w = c \xi_1,$$

where $c = 1/\|\xi_1\|_1$, and

$$D = \mathrm{diag}(|\lambda_1|, |\lambda_2|, \ldots, |\lambda_K|).$$

By Perron's theorem [1], $\lambda_1 > 0$ and all entries of $\xi_1$ are positive. Also, note that $\xi_1 \perp \xi_k$ for all $2 \leq k \leq K$. It follows

$$y^{(w)} = c D^{1/2} [\xi_1, \xi_2, \ldots, \xi_K]' \xi_1 = c \sqrt{\lambda_1} e_1,$$

so

$$y_0 = y^{(w)} / \|y^{(w)}\|_1 = e_1.$$

This justifies the first bullet point.

Consider the second bullet point. By definition,

$$\beta_i^{(w)} = \Omega w = c [\xi_1, \xi_2, \ldots, \xi_K] D^{1/2} J_{K,m} D^{1/2} [\xi_1, \xi_2, \ldots, \xi_K]' \xi_1 = c \lambda_1 \xi_1,$$

It follows

$$w' \widetilde{\Omega} w = w' \beta^{(w)} = c^2 \lambda_1 \xi_1' \xi_1 = c^2 \lambda_1.$$

At the same time, by definition,

$$\widetilde{\Omega} = YY' = \sum_{k=1}^{K} |\lambda_k| \xi_k \xi_k',$$

so

$$\widetilde{\Omega}(i,i) = \sum_{k=1}^{K} |\lambda_k| \xi_k^2(i).$$

Combining these,

$$\frac{|\beta_i^{(w)}|}{\sqrt{(w' \widetilde{\Omega} w) \widetilde{\Omega}(i,i)}} = \frac{c \lambda_1 \xi_1(i)}{\sqrt{c^2 \lambda_1 \sum_{k=1}^{K} |\lambda_k| \xi_k^2(i)}} = \frac{1}{\sqrt{1 + \sum_{k=1}^{K-1} (|\lambda_{k+1}|/\lambda_1) [\xi_{k+1}(i)/\xi_1(i)]^2}}.$$

Therefore, condition (2.10) of Theorem 2.3 reduces to

$$\frac{1}{\sqrt{1 + \sum_{k=1}^{K-1} (|\lambda_{k+1}|/\lambda_1) [\xi_{k+1}(i)/\xi_1(i)]^2}} \geq \sqrt{1 - 1/K}.$$

which is equivalent to (C.6). This completes the proof.

# D  Proof of Theorem 2.5

It is sufficient to show

- $a_m = (K-1)$ when $m = K - 1$.
- $a_m = 1$ when $m \leq K/2$.
- The NMF problem is solvable for $\Omega$ if $r_i' D_0 r_i \leq 1/[a_m(K-1)]$ for all $1 \leq i \leq n$.

Consider the first bullet point. By the definition of $\mathcal{Q}$, for all $Q \in \mathcal{Q}$, the first column of $Q$ is $K^{-1/2}e_1$. Therefore, when $m = (K-1)$, for any such $Q$, we have

$$Q = [e_0, Q^{(m)}].$$

By basic algebra,

$$2Q^{(m)}(Q^{(m)})' - I = 2(I_K - e_0 e_0') - I_K = I - (2/K)\mathbf{1}_K \mathbf{1}_K',$$

where the maximum entry of the matrix is $(K-2)/K$. By the definition, in this case, $a_m = 1 + K \cdot [(K-2)/K] = (K-1)$. This proves the first bullet point.

Consider the second bullet point. The goal is to show $a_m = 1$ for all $0 \leq m \leq K/2$. The case of $m = 0$ is trivial, so we suppose $m \geq 1$. First, we show

$$a_m \leq 1, \qquad \text{if } 1 \leq m \leq K/2. \tag{D.7}$$

For $1 \leq k \leq K/2$, let $h_k$ be the vector where

$$h_k(i) = \begin{cases} 1/\sqrt{2}, & i = 2k-1, \\ -1/\sqrt{2}, & i = 2k. \end{cases}$$

Construct $Q$ such that

- the first column is $e_0$,
- the last $m$ columns are $h_1, h_2, \ldots, h_m$, respectively.

For such a $Q$, by basic algebra, we have

- $Q \in \mathcal{Q}$.
- for the matrix $2Q^{(m)}(Q^{(m)})' - I_K$, none of the entries is positive.

The second item is true because, by construction, $2Q^{(m)}(Q^{(m)})'$ is a blockwise diagonal matrix, where all nonzero entries appear in $2 \times 2$ diagonal blocks of

$$\begin{bmatrix} 1 & -1 \\ -1 & 1 \end{bmatrix}.$$

This can be either checked by direct calculations or quoted from the proof of Lemma A.1. Combining these with the definition, $a_m \leq 1$, and (D.7) follows.

Next, we show

$$a_m \geq 1, \qquad \text{if } 1 \leq m \leq K/2. \tag{D.8}$$

We claim that in $m$ dimensional space, the maximum number of unit-norm vectors $r_i$ where the pairwise inner product are all (strictly) negative is no greater than $m + 1$. We prove this by mathematical induction. Note that this holds trivially when $m = 1$. Now, suppose this holds for $m \leq m_0$, we show the claim continues to hold for $m = m_0 + 1$. If the claim is not true for $m = m_0 + 1$, then there are $m_0 + 3$ different vectors

$$r_1, r_2, \ldots, r_{m_0+3}$$

of $(m_0 + 1)$ dimension where the pairwise inner product is strictly negative. Since all the pairwise inner products remain unchanged if we rotate these vectors by the same orthogonal matrix, we assume $r_{m_0+3} = (1, 0, \ldots, 0)'$ without loss of generality. Write

$$r_k = (a_k, s_k)', \qquad 1 \leq i \leq m_0 + 3.$$

where $a_k$ is the first entry of $r_k$ and $s_k$ is $m_0$-dimensional sub-vector of $r_k$. Since for all $1 \leq k \leq m_0 + 2$,

$$(r_k, r_{m_0+3}) < 0,$$

we must have

$$a_k < 0, \qquad \text{for all } 1 \leq k \leq m_0 + 2.$$

Therefore, for all $1 \leq k, \ell \leq K$ and $k \neq \ell$,

$$(s_k, s_\ell) = (r_k, r_\ell) - a_k a_\ell < 0.$$

It follows $s_1, s_2, \ldots, s_{m_0+2}$ (after scaled by a positive number) are unit-norm vectors of $m_0$-dimension where the pairwise inner product is strictly negative. This contradicts with our claim for the case of $m = m_0$. This completes the proof of mathematical induction.

Now suppose $a_m < 1$ for an $m \leq K/2$. By definition, there is a matrix $Q \in \mathcal{Q}$ such that

$$\text{all entries of } 2Q^{(m)}(Q^{(m)})' - I \text{ are strictly negative}, \tag{D.9}$$

where $Q^{(m)}$ is the sub-matrix of $Q$ consisting of the last $m$ columns. Write

$$Q^{(m)} = [r_1, r_2, \ldots, r_K]'.$$

For each $1 \leq k \leq K$, $r_k'$ the row $k$ of $Q^{(m)}$ and is $m$-dimensional. It follows from (D.9) that

$$(r_k, r_\ell) < 0, \qquad \text{for all } 1 \leq k, \ell \leq K, k \neq \ell.$$

By the above claim, this is only possible when $K \leq m + 1$. However, since $K \geq 3$ and $m \leq K/2$, so it is impossible that $K \leq m + 1$. The contradiction proves (D.8). Combining (D.7) and (D.8), $a_m = 1$ when $m \leq K/2$, which completes the proof of the second bullet point.

We now consider the last bullet point. Recall that $\lambda_k$ is the $k$-th eigenvalue of $\Omega$, $\xi_k$ is the corresponding eigenvector, and $\Xi = [\xi_1, \xi_2, \ldots, \xi_K]$. Define

$$\widetilde{Y} = \Xi \widetilde{D}^{1/2} = [\tilde{y}_1, \tilde{y}_2, \ldots, \tilde{y}_n]', \tag{D.10}$$

where

$$\widetilde{D} = \text{diag}(\lambda_1/a_m, |\lambda_2|, \ldots, |\lambda_K|).$$

Let $\widetilde{J}_{K,m}$ be the $K \times K$ diagonal matrix

$$\text{diag}(a_m, 1, \ldots, 1, -1, \ldots, -1). \tag{D.11}$$

That is, the first diagonal entry is $a_m$, the next $(K - m - 1)$ diagonal entries are $-1$, and all other diagonal entries are $-1$. It follows from spectral decomposition that

$$\Omega = \Xi \cdot \text{diag}(\lambda_1, \ldots, \lambda_K) \cdot \Xi' = \widetilde{Y}\widetilde{J}_{K,m}\widetilde{Y}'.$$

Therefore, we need to show is that, there is an orthogonal matrix $Q$ such that both matrix $\widetilde{Y}Q'$ and $Q\widetilde{J}_{K,m}Q'$ are non-negative.

To show the claim, note that by definition, there is a matrix $Q \in \mathcal{Q}$ such that

$$\max_{1 \leq i,j \leq K} \{H(i,j) : H = 2Q^{(m)}(Q^{(m)})' - I_K\} = (a_m - 1)/K, \tag{D.12}$$

Recall that by definition, the first column of any matrix $Q \in \mathcal{Q}$ is $e_0$. It follows

$$Q\widetilde{J}_{K,m}Q' = (a_m - 1)e_0 e_0' + I_K - 2Q^{(m)}(Q^{(m)})', \tag{D.13}$$

where since $e_0 = K^{-1}\mathbf{1}_K$, the right hand side is

$$\frac{(a_m - 1)}{K}\mathbf{1}_K\mathbf{1}_K' - [2Q^{(m)}(Q^{(m)})' - I_K]. \tag{D.14}$$

Combining (D.12)-(D.14) shows that the matrix $QJ_{K,m}Q'$ is non-negative.

At the same time, note that
$$Qe_1 = e_0,$$
where $e_1 = (1, 0, \ldots, 0)'$. Recall that $r_i \in \mathbb{R}^{K-1}$ and

$$r_i(k) = \xi_{k+1}(i)/\xi_1(i), \qquad 1 \leq k \leq K - 1, 1 \leq i \leq n.$$

Also, recall that $D_0 = \text{diag}(|\lambda_2|, \ldots, |\lambda_K|)$. By (D.10),

$$\tilde{y}_i \propto [1, r_i']\widetilde{D}^{1/2} = [\sqrt{\lambda_1/a_m}, r_i' D_0^{1/2}].$$

By basic algebra, for any $1 \leq i \leq n$,

$$\frac{|(\tilde{y}_i, e_1)|}{\|\tilde{y}_i\|} = \frac{1}{\sqrt{1 + (a_m/\lambda_1)r_i' D_0 r_i}},$$

where by the condition of Theorem 2.5,

$$a_m r_i' D_0 r_i \leq \lambda_1/(K-1). \tag{D.15}$$

It follows that

$$\frac{|(\tilde{y}_i, e_1|}{\|\tilde{y}_i\|} \geq \sqrt{1-1/K}.$$

Combining this with $Qe_1 = e_0$,

$$\frac{|(Q\tilde{y}_i, e_0)|}{\|Q\tilde{y}_i\|} = \frac{|(Q\tilde{y}_i, Qe_1)|}{\|Q\tilde{y}_i\|} = \frac{|(\tilde{y}_i, e_1)|}{\|\tilde{y}_i\|} \geq \sqrt{1-1/K}.$$

By Lemma A.2, $Q\tilde{y}_i$ falls in the first orthant. Since this holds for all $1 \leq i \leq n$, $\widetilde{Y}Q'$ is non-negative. This finishes the proof of Theorem 2.4.

## E  Proof of Lemma 3.1 and Lemma 3.2

Recall that $D = \text{diag}(|\lambda_1|, |\lambda_2|, \ldots, |\lambda_K|)$ and $D_0 = \text{diag}(|\lambda_2|, \ldots, |\lambda_K|)$. Similarly, define

$$H = \text{diag}(\lambda_1, \lambda_2, \ldots, \lambda_K), \qquad \text{and} \qquad H_0 = \text{diag}(\lambda_2, \lambda_3, \ldots, \lambda_K).$$

It is sufficient to show

- If $\lambda_1(G) \leq c_0 \lambda_K(G)$ for a constant $c_0 > 0$, then as $n \to \infty$, the two conditions of $\max_{2 \leq k \leq K}\{|\lambda_k/\lambda_1|\} \to 0$ and $\max_{2 \leq k \leq K}\{|\lambda_k(P)/\lambda_1(P)|\} \to 0$ are equivalent.
- If $\lambda_1(G) \leq c_0 \lambda_K(G)$, then for all $1 \leq i \leq n$, $\|r_i\| \leq CM(\Omega)$.
- $B = \text{diag}(b_1)[\mathbf{1}_K, V]$, $P = BHB'$, $P(k,k) = b_1^2(k)/[\lambda_1 + v_k' H_0 v_k]$, $1 \leq k \leq K$, and $b_1$ is an eigenvector of $PG$ with $\lambda_1$ being the corresponding eigenvalue.
- If $(Y, P)$ are non-negative, then first, $PG$ is an irreducible non-negative matrix and $b_1$ is the Perron vector (so all entries of $b_1$ are strictly positive). Second, all rows of $r_i$ lives with a simplex with $v_1, v_2, \ldots, v_K$ being the vertices. Last, if $\lambda_1(G) \leq c_0 \lambda_K(G)$, then $M(\Omega) \leq \max_{1 \leq k \leq K}\{\|b_1\|/b_1(k)\}$.

Consider the first bullet point. It is sufficient to show

- if $\max_{2 \leq k \leq K}\{|\lambda_k/\lambda_1|\} \to 0$ then $\max_{2 \leq k \leq K}\{|\lambda_k(P)/\lambda_1(P)|\} \to 0$,
- if $\max_{2 \leq k \leq K}\{|\lambda_k(P)/\lambda_1(P)|\} \to 0$ then $\max_{2 \leq k \leq K}\{|\lambda_k/\lambda_1|\} \to 0$.

Consider the first item. Recall that $B = G^{-1/2}Q$ for a $K \times K$ orthogonal matrix $Q$, and $G$ is a positive definite matrix satisfying

$$\lambda_1(G) \leq C\lambda_K(G). \tag{E.16}$$

Recall that $B = [b_1, b_2, \ldots, b_K]$. Write $Q = [q_1, q_2, \ldots, q_K]$ and let

$$G = UDU'$$

be the spectral decomposition of $G$, where $U \in \mathbb{R}^{K,K}$ is orthogonal and

$$D = \text{diag}(\lambda_1(G), \lambda_2(G), \ldots, \lambda_K(G)).$$

It follows

$$B = G^{-1/2}Q = UD^{-1/2}U'Q, \qquad b_1 = UD^{-1/2}U'q_1.$$

Therefore, by (E.16),

$$\|B\| = \|G^{-1/2}Q\| = \|G^{-1/2}\| \leq 1/\sqrt{\lambda_K(G)}, \tag{E.17}$$

and for $1 \leq k \leq K$,

$$\|b_k\|^2 = \|UD^{-1/2}U'q_k\|^2 = q_k'UD^{-1}U'q_k \asymp (1/\lambda_K(G))\|U'q_k\| = 1/\lambda_K(G). \tag{E.18}$$

Now, using the first bullet point, we write

$$P = BHB' = B \cdot \text{diag}(\lambda_1, \ldots, \lambda_K) \cdot B = P_0 + P_1,$$

where
$$P_0 = B \cdot \mathrm{diag}(\lambda_1, 0, \ldots, 0) \cdot B', \qquad P_1 = B \cdot \mathrm{diag}(0, \lambda_2, \ldots, \lambda_K) \cdot B'.$$

Note that
$$P_0 = \lambda_1 b_1 b_1',$$

where $\lambda_1 > 0$. Therefore,
$$\lambda_k(P_0) = \begin{cases} \lambda_1 \|b_1\|^2, & k = 1, \\ 0, & 2 \le k \le K. \end{cases} \tag{E.19}$$

By Weyl's theorem [1], for any $1 \le k \le K$,
$$|\lambda_k(P) - \lambda_k(P_0)| \le \|P_1\|,$$

where by (E.18),
$$\|P_1\| \le \|B\| \cdot \max_{\{2 \le k \le K\}} \{|\lambda_k|\}.$$

Therefore,
$$|\lambda_k(P) - \lambda_k(P_0)| \le (1/\lambda_K(G)) \cdot \max_{\{2 \le k \le K\}} \{|\lambda_k|\}. \tag{E.20}$$

Combining (E.19)-(E.20) with (E.18),
$$\lambda_1(P) \ge \lambda_1 \|b_1\|^2 - (1/\lambda_K(G)) \cdot \max_{\{2 \le k \le K\}} \{|\lambda_k|\} \ge (1/\lambda_K(G))[C\lambda_1 - \max_{\{2 \le k \le K\}} \{|\lambda_k|\}],$$

and
$$|\lambda_k(P)| \le (1/\lambda_K(G)) \cdot \max_{\{2 \le k \le K\}} \{|\lambda_k|\}, \qquad 2 \le k \le K.$$

Therefore, if
$$(1/\lambda_1) \max_{\{2 \le k \le K\}} \{|\lambda_k|\} \to 0,$$

then $\lambda_1(P) > 0$ and
$$(1/\lambda_1(P)) \max_{\{2 \le k \le K\}} \{|\lambda_k(P)|\} \to 0.$$

This proves the first item.

The proof of the second item is similar, so we keep it short. Let $P = UDU'$ be the spectral decomposition of $P$, where $U$ is a $K \times K$ orthogonal matrix, and
$$D = \mathrm{diag}(\lambda_1(P), \ldots, \lambda_2(P)).$$

Combining these with $P = BHB'$ and $B = G^{-1/2}Q$,
$$H = B^{-1}U'DU'(B^{-1})' = (Q'G^{1/2}U')D(UG^{1/2}Q).$$

Recall that
$$H = \mathrm{diag}(\lambda_1, \ldots, \lambda_K).$$

It follows
$$\mathrm{diag}(\lambda_1, \ldots, \lambda_K) = (Q'G^{1/2}U') \cdot \mathrm{diag}(\lambda_1(P), \ldots, \lambda_K(P)) \cdot (UG^{1/2}Q).$$

The remaining part of the proof is similar, so we skip it. This completes the proof of the first bullet point.

Consider the second bullet point. Recall that
$$\Xi = [\xi_1, \ldots, \xi_K] = YB, \qquad r_i(k) = \xi_{k+1}(i)/\xi_1(k), 1 \le k \le K-1, 1 \le i \le n,$$

and that $Y = [y_1, y_2, \ldots, y_n]'$ and $B = [b_1, b_2, \ldots, b_n]$. We have
$$r_i(k) = y_i'b_{k+1}/y_i'b_1.$$

By definition, $y_i'b_1 = \|y_i\|\|b_1\|\cos(\alpha_i)$, so
$$|r_i(k)| \le \frac{\|y_i\|\|b_{k+1}\|}{\|y_i\|\|b_1\||\cos(\alpha_i)|} = \frac{1}{|\cos(\alpha_i)|}\frac{\|b_{k+1}\|}{\|b_1\|}.$$

By (E.18),
$$\frac{\|b_{k+1}\|}{\|b_1\|} \asymp 1.$$
Therefore,
$$|r_i(k)| \le C/|\cos(\alpha_i)| \le CM(\Omega).$$
Since $K$ is finite, the claim follows. This proves the second bullet point.

Consider the third bullet point. The first item follows directly by definition of $V$. For the second item, recall that
$$\Omega = \Xi H \Xi' = YPY', \qquad \text{and} \qquad \Xi = YB.$$
Combining these gives
$$YBHB'Y' = YPY',$$
and the claim follows since $Y$ is full-rank by our assumption.

For the third item, we combine the first two items, and it follows that
$$P = BHB' = \text{diag}(b_1)[\mathbf{1}_K, V]H[\mathbf{1}_K, V]'\text{diag}(b_1) = \text{diag}(b_1)[\lambda_1 \mathbf{1}_K \mathbf{1}_K' + VH_0V']\text{diag}(b_1).$$
Recall that
$$V = [v_1, v_2, \ldots, v_K]'.$$
For any $1 \le k \le K$, comparing the $k$-th diagonal entry of the two matrices, $P$ and $\lambda_1 b_1 b_1' + VH_0V'$, it follows that
$$P(k, k) = b_1^2(k)[\lambda_1 + v_k'H_0v_k].$$
This proves the third item.

Consider the last item. By basic algebra, the set of all nonzero eigenvalues of the matrix $YPY'$ are the same as the set of all nonzeor eigenvalues of $PY'Y$ or $PG$. Since $PG$ is non-singular, the eigenvalues are $\lambda_1, \ldots, \lambda_K$. By $\Xi = [\xi_1, \ldots, \xi_K] = YB$ and $B = [b_1, b_2, \ldots, b_K]$, we have
$$\xi_k = Yb_k.$$
Since $\xi_1$ is the eigenvector corresponding to $\lambda_k$, so on one hand,
$$\lambda_k \xi_k = \lambda_k Yb_k,$$
and on the other hand,
$$\lambda_k \xi_k = \Omega \xi_1 = YPY'\xi_k = YPY'Yb_k = YPGb_k.$$
Combining these and noting that $Y$ is full rank,
$$PGb_k = \lambda_k b_k.$$
Therefore, $b_1, b_2, \ldots, b_K$ are singular vectors corresponding to $\lambda_1, \ldots, \lambda_K$. This proves the last item and completes the proof of the second bullet point.

Consider the last bullet point. Consider the first item first. Since $Y$ and $P$ are non-negative, $PG$ is non-negative. Also, the matrix $PG$ is also irreducible since $\Omega$ is irreducible. Since $b_1$ is the Perron vector of $PG$, all of entries are strictly positive.

Consider the second item. Fix $1 \le i \le n$. Since $\Omega$ is irreducible, $y_i \ne 0$. Introduce a weigh vector $w_i \in \mathbb{R}^K$ by $w_i(k) = y_i(k)b_1(k)/(y_i'b_1)$, $1 \le k \le K$. Recall that $V(j, k) = b_{k+1}(j)/b_1(j)$, $1 \le k \le K-1, 1 \le j \le K$. We have
$$r_i(k) = \frac{y_i'b_{k+1}}{y_i'b_1} = \sum_{j=1}^{K} w_i(j)(b_{k+1}(j)/b_1(j)) = \sum_{j=1}^{K} w_i(j)V(j, k).$$
Therefore,
$$r_i = w_i'V,$$
and $r_i$ is a convex linear combination of the $K$ rows of $V$. This says $r_i$ fall within the simplex with $v_1, v_2, \ldots, v_K$ being the vertices. This proves the second item.

Consider the last item. Note that for any $1 \leq i \leq n$, $y_i \neq 0$. Otherwise, if $y_i = 0$ for some $i$, then $\Omega(i,j) = \Omega(j,i)$ for all $j$, and $\Omega$ is reducible. Now,

$$\cos(\alpha_i) = \frac{(y_i, b_1)}{\|y_i\| \|b_1\|}.$$

Without loss of generality, assume $\|y_i\|_1 = 1$ since $y_i$ is non-negative vector. It follows

$$\|y_i\| = \sqrt{\sum_{k=1}^{K} y_i^2(k)} \leq \sqrt{\|y_i\|_1} = 1.$$

Therefore,

$$|\cos(\alpha_i)| \geq \min_{\{1 \leq k \leq K\}} \{b_1(k)\} \|y_i\|_1 / (\|y_i\| \|b_1\|) \geq \min_{\{1 \leq k \leq K\}} \{b_1(k)/\|b_1\|\}.$$

This proves the last item and completes the proof of the last bullet point.

## F    Comments on how to find $(Q, y_0)$ in Theorem 2.1 numerically

Given $\Omega = Y J_{K,m} Y'$ as in Theorem 2.1, an interesting question is how to construct a $Q$ numerically when condition (2.9) of Theorem 2.1 holds for some $y_0$. The proof of Theorem 2.1 relies on a specific construction of Q (see the proof of Theorem 2.1, especially Lemma A.3 for details) as follows.

- Let $Q_1$ be the $K \times K$ orthogonal matrix with the form of

$$Q_1 = \begin{bmatrix} Q_0 & 0 \\ 0 & I_m \end{bmatrix}, \qquad \text{the first row of } Q_0 \text{ is } (y_0(1), \ldots, y_0(K-m)).$$

- Let $Q_2$ be the $K \times K$ orthogonal matrix with the form of

$$Q_1 = [e_0, q_1, \ldots, q_{K-m-1}, h_1, h_2, \ldots, h_m], \qquad e_0 = (1/\sqrt{K})(1, 1, \ldots, 1)',$$

  where for $k = 1, 2, \ldots, m$,

$$h_k(i) = \begin{cases} 1/\sqrt{2}, & i = 2k-1, \\ -1/\sqrt{2}, & i = 2k, \\ 0, & \text{otherwise.} \end{cases}$$

- Let $Q = Q_2 Q_1$.

Therefore, numerically, all remains is to decide the remaining rows of $Q_0$ and the remaining columns of $Q_2$, both can be solved quickly by basic algebra, since $K$ is usually small.

A related question is how to check whether there is a vector $y_0 \in \mathcal{S}_{K,m}$ such that condition (2.9) of Theorem 2.1 holds. Recalling that $Y = [y_1, y_2, \ldots, y_n]'$, possible candidate are $y_i^*$, $1 \leq i \leq n$, where

$$y_i^*(k) = \begin{cases} y_i(k), & 1 \leq k \leq K-m, \\ 0, & k > K-m. \end{cases}$$

The $y_i^*$ constructed this way belong to $\mathcal{S}_{K,m}$. Now, without loss of generality, assume $y_i^* \neq 0$ for all $1 \leq i \leq n$. For each $1 \leq s \leq n$, let

$$U_s = \max_{1 \leq i \leq n} \{|(y_s^*, y_i)| / (\|y_i\| \|y_s^*\|)\}.$$

If $\max_{1 \leq s \leq n} \{U_s\} > \sqrt{1 - 1/K}$, then condition (2.9) does not hold and it is unclear if the NMF problem is solvable for $\Omega$. If $\max_{1 \leq s \leq n} \{U_s\} > \sqrt{1 - 1/K}$, then condition (2.9) holds with $y_0 = y_{\hat{k}}^*$, and the NMF problem is solvable.

## References

[1]  Roger A. Horn and Charles R. Johnson. *Matrix Analysis*. Cambridge University Press, 2nd edition, 2013.