# OpenReview forum: "A sharp NMF result with applications in network modeling  "
_NeurIPS.cc/2022/Conference — NeurIPS 2022 Accept_

### Official Review · Reviewer_1VWq · 2022-07-03

**Rating:** 6
**Confidence:** 4
**Soundness:** 3 good
**Presentation:** 3 good
**Contribution:** 4 excellent

**Summary:**

In this paper, the authors discuss when a square symmetric non-negative matrix with $m$ negative eigenvalues allows a non-negative factorization of the form $ZPZ'$. The authors' main result is Theorem 2.1 where they show a set of sufficient conditions under which the NMF problem is solvable. The authors also extend the results for some special cases.

**Questions:**

1) Here, the paper discusses the solvability of the NMF problem. However, of equal importance is the question of when the solution is unique and can be recovered by an algorithm. The paper does not answer these questions for problem 1.1 when the problem is known to be solvable.

2) How did J_{k,m} still come up in the spectral decomposition of \Omega? \lambda and \Zeta are eigenvalues and eigenvectors of the symmetric matrix \Omega; so \Omega should just be \Zeta D\Zeta' , right?

Minor:

1) Why use such an uncommon notation for the inner product that is so confusing?
2) How is the matrix of entry-wise ratio relevant (L246) and what do its entries correspond to? Also, what is R in 2.13?


**Limitations:**

Yes

**Strengths And Weaknesses:**

The paper is well-written and also provides nice examples to motivate the problem. However, I feel that there are several questions that are not answered properly and are mentioned below.

---

> ### Author Response · Authors · 2022-08-01
> **Response to Review 1VWq**
>
> We would like to thank you for your time and valuable comments.
> We are especially glad that you recognize the main contributions
> of our paper, and think our paper as well-written and well-motivated.
> We have tried our best to address your comments
> and prepared a revised version and a point-to-point response below.
>
>
>
> 1a. Response to comment 1 (part 1 on uniqueness).  When the NMF is solvable, it is known that the solution is usually not unique without a proper regularity condition (e.g., \cite{DonohoNMF}). In our setting, once we can factor $\Omega$ by $\Omega = \Theta \Pi P \Pi' \Theta$ for some non-negative matrices $(\Theta, \Pi, P)$ as in equation (1.5),  the factorization is unique if (a) for each $1 \leq k \leq K$, there is at least one $i$ such that $\pi_j = e_k$, where $e_1, \ldots, e_K$ are the standard Euclidean basis vectors of $\mathbb{R}^K$, and (b) all diagonal entries of $P$ are $1$.  For the proof, see \cite{JKL2019} for example.  In response, we have added Remark 4 near the end of Section 2 (marked in red).
>
>
>
> 1b. Response to comment 1 (part 2 on algorithm for NMF).  This is an interesting question. In response,
> we have added Remark 5 at the end of Section 2 (marked in red), and also added a new section, Section F, in the supplement (the discussion is almost a page, so we put it in the supplement). For details, please read with Remark 5 and Section F of the supplement.
>
> 2. Response to comment 2 (why some entries of $J_{K, m}$ are still negative):  Thanks for the comments. Recall that by SVD,
> \begin{equation}
> \Omega = \Xi  \mathrm{diag}(\lambda_1, …, \lambda_K) \Xi’, \qquad \mbox{where $\Xi = [\xi_1, \ldots, \xi_K]$}.
> \end{equation}
> You are right if all eigenvalues of $\Omega$ are positive. But when some of the eigenvalues of $\Omega$ are negative, we have
> \begin{equation}
> \Omega = \Xi  \mathrm{diag}(\lambda_1, …, \lambda_K) \Xi’, \qquad \mbox{where $\Xi = [\xi_1, \ldots, \xi_K]$}.
> \end{equation}
> In this case, we can not take the square root of the matrix $\mathrm{diag}(\lambda_1, …, \lambda_K)$ directly.  Instead, we introduce a positive diagonal  matrix by
> \begin{equation}
> D = \mathrm{diag}( |\lambda_1|, …, |\lambda_K|).
> \end{equation}
> Therefore,
> \begin{equation}
> \Omega = \Xi D^{1/2} J_{K, m} D^{1/2}  \Xi,
> \end{equation}
> and the matrix $J_{K, m}$ is still there.
>
>
>
>
> 3. Response to minor comment 1 (why use an unconventional notation for the inner product): Thanks and we think what you meant is that why not use $<x,y>$ instead of $(x,y)$  as the notation for the inner product.
> Both notations are frequently used, and we have also explained what $(x,y)$ stands for in
> section 1.4.
>
>
>
> 4. Response to minor comment 2 (entrywise ratio matrix): Thanks for the comments. The R matrix in equation (2.13) was introduced right above equation (2.13). Let $\mathrm{diag}(\xi_1)$ be the $n \times n$ diagonal matrix where the $j$-th diagonal entry is $\xi_1(j)$, $1 \leq j \leq n$.   It is seen that
> \begin{equation}
> [\xi_1, \xi_2, …, \xi_K] = \mathrm{diag}(\xi_1) [{\bf 1}_n, R],  \qquad \mbox{where ${\bf 1}_n$ is the vector of all ones},
> \end{equation}
> and so
> \begin{equation}
> \Omega = \mathrm{diag}(\xi_1) [1_n, R] D [1_n, R]’ \mathrm{diag}(\xi_1).
> \end{equation}
> By Perron’s theorem, all entries of $\xi_1$  are strictly positive. Therefore, the NMF problem is solvable for $\Omega$  if and only if the NMF problem is solvable for
> \begin{equation}
> [{\bf 1}_n, R] D [{\bf 1}_n, R]’.
> \end{equation}
> Compared with $\Omega$, this matrix is much simper. This is why we choose to state Theorem 2.4 using the rows of $R$
> instead of those of $\Omega$.

---

### Official Review · Reviewer_61UN · 2022-07-12

**Rating:** 6
**Confidence:** 3
**Soundness:** 3 good
**Presentation:** 3 good
**Contribution:** 3 good

**Summary:**

The paper deals with the problem of symmetric non-negative matrix factorization (NMF) for a low-rank matrix, when some of the eigenvalues of the matrix are negative. The paper provides conditions under which a symmetric NMF solution exists for a matrix of rank K with negative eigenvalues. The main theoretical results are divided into two main theorems, one for m<K/2 and another for general K. The paper also provides an algorithm for symmetric NMF for degree-corrected mixed-membership block model (DCMM).

**Questions:**

Suggestion:
1. Regarding section 4, it would be better to either put in Supplementary or give the theoretical support behind the estimated conditions.

**Limitations:**

None noted.

**Strengths And Weaknesses:**

Strength:
1. The theoretical results (Theorem 2.1-2.5) on the conditions under which a symmetric NMF solution exists for a matrix of rank K with negative eigenvalues are the main strengths of the paper, as the results need some technical innovation.
Weakness:
1. The algorithm of NMF for DCMM is provided a bit lightly and without theoretical support, thus section 4 becomes quite antithetical to the tone of the rest of the paper.
2. The paper also misses some relevant references, like,
(i) Anandkumar, Animashree, Ge, Rong, Hsu, Daniel J, and Kakade, Sham M. A tensor approach to learning mixed membership community models. Journal of Machine Learning Research, 15(1):2239–2312, 2014.
(ii) Mao, X., Sarkar, P. and Chakrabarti, D., 2017, July. On mixed memberships and symmetric nonnegative matrix factorizations. In International Conference on Machine Learning (pp. 2324-2333). PMLR.

---

> ### Author Response · Authors · 2022-08-01
> **Response to Review 61UN**
>
> We would like to thank you for your time and valuable comments.
> We are especially glad that you recognize  the main strength and
> contributions of our paper. We have tried our best to address your comments
> and prepared a revised version and a point-to-point response below.
>
>
> 1. Response to comment 1 (on Section 4):  Thanks for your comments. Section 4 provides a real data example to complement
> with the theory. The main focus of the paper is that, given an $\Omega$, study when the
> NMF problem is solvable. For numerical study, many existing works in this area use a simulated
> $\Omega$ matrix, but we think that using the $\Omega$ matrix from a real network is more
> interesting. The challenge is however that $\Omega$ is unknown in real applications.
> Therefore, we have to develop a new algorithm for estimating $\Omega$. How to analyze
> this algorithm is interesting but is not our main focus, so we leave it to the future.
> I think some readers would desire to see a real data example so we prefer to keep
> Section 4 in the main text.
>
>
>
> 2. Response to comment 2 (on missed references):  Thanks for pointing out these very interesting references. We have added them to the paper.

---

### Official Review · Reviewer_h5Vj · 2022-07-13

**Rating:** 5
**Confidence:** 2
**Soundness:** 3 good
**Presentation:** 3 good
**Contribution:** 3 good

**Summary:**

This paper considers a particular case of the NMF problem, along with its application to network modelling.
The main contributions of the paper is two-fold. On one hand, the authors put forth several new results on the symmetric case of the NMF problem, further generalizing previous lines of work that considered special cases of the current formulation from the paper. On the other hand, as a secondary contribution, the authors provide a both quantitative and qualitative discussion on use-cases of different rank-K network models.


**Questions:**

Perhaps not a good idea to use acronyms in the abstract without first explaining them - most readers of the abstract would not be familiar with DCMM, so a brief comment would be appropriate.

What about the asymetric case? Can a brief comment be made on it, in light of the results in this paper?

Could clarify what are “pure” nodes in Remark 1, line 82 - as it might not be clear to everyone outside of the SBM community.

What about extension to a directed stochastic block model? (say, where the input adjacency matrix is a skew symmetric matrix). And somewhat related to this, what about the complex NMF?

Why exactly is the case m < K/2 - “probably the most interesting case in practice”?

The notation J_{K,m} is a bit cumbersome to follow at places, perhaps a notation can be used without the use of subscripts throughout, to improve readability.

In Section 3, the authors repeatedly refer to “network analysis” applications as if it was a particular problem/task - but it should be made clear at the beginning what is the exact problem to be adressed.

Are there any implications of the results concerning the ability to derive guarantees of SBM/community detection models in the very sparse regime? (one where the edge density in the graph is required to be above (log n) / n and extra effort is required for regularize appropriately before procedding with a spectral approach).

The authors could comment more on the converse of Thm 2.4 when first mentioned.

The Figure in page 9 should have a bare minimum to axis labeling and sub/captions.

Section 4 could make it clear what type of insights one can obtain with the NMF-based approach which perhaps it cannot otherwise.


Typos:
314: all rows of r_i lives with
357 (n,K) = (1,222, 2) - confusing at first

**Strengths And Weaknesses:**

The paper is fairly original in its inception, and provides a generation of previous results, with previous results being recovered from the current formulation. It provided a very good motivation for why one might care to consider an NMF approach to analyze network problems. The paper is quite dense, and a bit difficult to follow at places, also due to several omitted proofs and results called upon from other works, and not always with enough context. The authors could consider improving this aspect.

---

> ### Author Response · Authors · 2022-08-01
> **Response to Review h5Vj (part 1)**
>
> We would like to thank you for your time and valuable comments.
> We are especially glad that you think our results are new and
> that we have made a two-fold contribution. We have tried our best to address your comments
> and prepared a revised version and a point-to-point response below.
>
>
>
> 1. Response to comment 1 (do not use acronym DCMM in the abstract): Great comments. We have made the changes.
>
> 2. Response to comment 2 and 4:   These are great comments. The settings you suggested (asymmetric, directed SBM,  complex NMF) are very interesting.  These settings are very different from the setting considered here and deserve a careful study in a future paper.  Here, we present a straightforward extension of Theorem 2.1 to the asymmetric case.   By SVD, for an $n \times p$ (entry-wise) positive matrix Omega with rank K, we can write $\Omega = Y Z’$ for an $n \times K$ matrix $Y$ and an $p \times K$ matrix $Z$. Let $y_i'$ be $i$-th row of $Y$ and $z_j’$ be the $j$-th row of Z, respectively. If there is a $K$ dimensional vector $y_0$ such that for all $i$ and $j$,
> \begin{equation}
> |(y_i, y_0)| / \|y_i\| \geq \sqrt{1 - 1/K}, \qquad \mbox{and} \qquad |(z_j, y_0)|/\|z_j\| \geq \sqrt{1 - 1/K},
> \end{equation}
> then we can find a $K \times K$ orthogonal matrix $Q$ which rotates all rows of $Y$ and $Z$ to the first orthant simultaneously. In this case, the NMF problem is solvable. In response, we have added some references and a few sentences to the end of paragraph right below Theorem 2.1 (marked in red).
>
>
> 3. Response to comment 3 on pure nodes: thanks and we have made the changes (see two places marked in red on Page 3).
>
> 4. Response to comment 4: see item 2 above.
>
> 5. Response to comment 5 (why $m < K/2$ is the most interesting case in practice):
> This is a great comment. We think you question is about the first sentence of Section 2.4.  We meant that we have $m < K/2$  for most real networks we have analyzed.  We have
>  checked the eigenvalues of the adjacency matrices of more than a handful of frequently seen network data sets.  For most of them, we have $m < K/2$.  In response,  we have rephrased the sentence as “which is the case that is most frequently found for real networks).
>
> 6. Response to comment 6 (the notation $J_{K,m}$): Thanks and you certainly have a good point. However,  the subscripts are used to stress that the matrix is a $K \times K$ diagonal matrix where among the $K$ diagonal entries,  the last $m$ of them are $-1$ and others are $1$. It is hard  to find a simpler notation.
>
> 7. Response to comment 7 (what specific problem in network analysis to study in the beginning of Section 3):
> I think what you mean is that the term of network analysis is too broad and too abstract,
> and it is better to make it more specific.  This is a great point. In response,
> in the first paragraph of Section 3, we have changed Network analysis to "Network analysis (e.g., community detection, membership estimation, link prediction)".  We hope this makes it more specific.

---

> ### Author Response · Authors · 2022-08-01
> **Response to review h5Vj (part 2)**
>
>  8.  Response to comment 8 (connections to community detection):  This is a great comment.
> To develop an approach to community detection, we desire to have a more specific model (a model
> that has many structures we can exploit).  Our paper develops an approach to checking when we can rewrite a broader model (the rank-$K$ model) to a more specific model (the DCMM model), and so it is helpful for
> community detection for it ensures us that there are more structures in the model which we can exploit.
>
> For example,  let $\xi_k$ and $V = [v_1, v_2, \ldots, v_K]'$ be as in Section 3 (near equation (3.15)), and
> let $R$ be the $n \times (K-1)$ matrix satisfying $R(i, k) = \xi_{k+1}(i) / \xi_1(i)$, $1 \leq k \leq K-1, 1 \leq i \leq n$. If we view each row of $R$ as a point in $\mathbb{R}^{K-1}$, then there is a simplex in $\mathbb{R}^{K-1}$  with vertices $v_1, v_2, \ldots, v_K$ where row $i$ of $R$ falls on one of the vertex if node $i$ is pure and $R$ falls in the interior of the simplex otherwise. Such a result is helpful in obtaining a more accurate
> estimate for the $\Omega$ matrix (this is a hard problem and the main reason is that the $\Pi$ matrix is latent and it is hard to estimate; fortunately, the simplex structure suggested an efficient approach to estimating $\Pi$ (e.g., \cite{JKL2017, JKLW2022})).  For a general rank-$K$ network model, we do not have the simplex structure, so it remains unclear how to develop an accurate estimate for the $\Omega$ matrix (we can use ordinary spectral decomposition but the resultant estimate is frequently unsatisfactory).
> In response, we have added a few sentences (marked in read) to the first paragraph of Section 5.
>
>
> 9.  Response to comment 9 (on the converse of Theorem 2.4):
> Thanks for the comments. We think what you suggested is to comment on whether condition (2.14) is
> sufficient and necessary. As in most works on NMF (e.g., the book by Shaked-Monderer and Berman cited in our paper), our goal is to find conditions that are easy to check and sufficient. Such conditions are usually not necessary, but necessary conditions are usually messy and hard to check (e.g., see the book by  Shaked-Monderer and Berman).
> We have added some sentences right below Theorem 2.4 (marked in red).
>
> 10. Response to comment 10 (figure on Page 9):   Thanks for your comments. In the previous version, the caption is omitted for space limit.  We have now added a caption. See Page 10 (marked in red).
>
> 11. Response to comment 11 (in Section 4, mention what insight NMF can offer while other approaches can not): This is a very helpful comment.  In many recent works on network analysis, we frequently assume that a DCMM model holds for the settings at hand, but we rarely checked if such an assumption is valid.
> Our NMF results provide an approach to checking whether the network satisfies DCMM model.
> We have added a few sentences to the end of Section 4 (marked in red).
>
>
> 12. Response to comment 12 (typos): thanks and they are corrected.

---

### Meta-Review · Area_Chair_cmT6 · 2022-08-30

**Recommendation:** Accept
**Confidence:** Less certain

**Metareview:**

This was a borderline paper, which fell just above the bar for acceptance. The reviewers felt the work was interesting and original, although perhaps the problem studied is a bit niche for NeurIPS.

**Award:**

No

---

### Decision · Program_Chairs · 2022-09-14

Accept